# Exosomal cargo including microRNA regulates sensory neuron to macrophage communication after nerve trauma

Raffaele Simeoli[1], Karli Montague [1], Hefin R. Jones[2], Laura Castaldi [3], David Chambers[1], Jayne H. Kelleher[1], Valentina Vacca [1,4], Thomas Pitcher[1], John Grist[1], Hadil Al-Ahdal [5], Liang-Fong Wong [5], Mauro Perretti [2], Johnathan Lai[6], Peter Mouritzen[6], Paul Heppenstall[3] & Marzia Malcangio [1]

Following peripheral axon injury, dysregulation of non-coding microRNAs (miRs) occurs in dorsal root ganglia (DRG) sensory neurons. Here we show that DRG neuron cell bodies release extracellular vesicles, including exosomes containing miRs, upon activity. We demonstrate that miR-21-5p is released in the exosomal fraction of cultured DRG following capsaicin activation of TRPV1 receptors. Pure sensory neuron-derived exosomes released by capsaicin are readily phagocytosed by macrophages in which an increase in miR-21-5p expression promotes a pro-inflammatory phenotype. After nerve injury in mice, miR-21-5p is upregulated in DRG neurons and both intrathecal delivery of a miR-21-5p antagomir and conditional deletion of miR-21 in sensory neurons reduce neuropathic hypersensitivity as well as the extent of inflammatory macrophage recruitment in the DRG. We suggest that upregulation and release of miR-21 contribute to sensory neuron–macrophage communication after damage to the peripheral nerve.

[1] Wolfson Centre for Age Related Diseases, King's College London, London SE1 1UL, UK. [2] The William Harvey Research Institute, Barts and The London School of Medicine, Queen Mary University of London, London EC1M 6BQ, UK. [3] EMBL Monterotondo, Via Ramarini 32, 00016 Monterotondo, Italy. [4] Institute of Cell Biology and Neurobiology, National Research Council and IRCCS Fondazione Santa Lucia, 00143 Rome, Italy. [5] School of Clinical Sciences, Medical Science Building, University of Bristol, Bristol BS8 1TD, UK. [6] Exiqon A/S, Skelstedet 16, 2950 Vedbaek, Denmark. Correspondence and requests for materials should be addressed to M.M. (email: marzia.malcangio@kcl.ac.uk)

Neuropathic pain is a debilitating condition and the efficacy of current treatment strategies, which include opioids and anticonvulsants, is limited by the extensive side effect profiles observed in patients[1]. Thus, there is a necessity for novel mechanisms and therapeutic targets to be identified. Compelling evidence supports a critical role of immune cells in the mechanisms underlying neuropathic pain at the site of nerve damage in the periphery, in the dorsal root ganglia (DRG), and in the dorsal horn of the spinal cord[2]. At the site of injury and in the

DRG, monocytes/macrophages infiltrate in response to chemokines produced by Schwann cells and satellite cells. Proinflammatory macrophages release mediators such as cytokines and chemokines, which activate the vascular endothelium and alter the sensory transduction properties of nociceptive axons and cell bodies, causing continual activity (peripheral sensitization)[3, 4]. In the spinal cord, microglia proliferate, change their morphology, undergo changes in gene expression, and release pro-nociceptive mediators, which can sensitize neurons and contribute to central

**Fig. 1** Expression of miR-21 is increased in DRG neurons following spared nerve injury. **a–d** Upregulation of miR-21 expression detected by fluorescence in situ hybridization (FISH) in ipsilateral L5 DRG neurons 7 days after SNI compared to sham injury and contralateral DRG neurons. Scale bar = 100 μm. **e** Quantification of miR-21[+] neurons in L4/5 DRG. **f, g** Immunostaining for large-diameter DRG neurons (NF-200, red) and FISH for miR-21 (green) in sham and SNI ipsilateral L5 DRG. Scale bar = 100 μm. **h** Quantification of large cell bodies NF-200[+] neurons that also express miR-21 in L4/5 DRG. **i, j** Immunostaining of small-diameter DRG neurons (CGRP, red) and FISH for miR-21 (green) in sham and SNI ipsilateral L5 DRG. Scale bar = 100 μm. **k** Quantification of CGRP[+] neurons that also express miR-21 in L4/5 DRG. **l–o** Immunostaining of macrophages (F4/80[+] cells, red), FISH for miR-21 (green), and nuclei (4′,6-diamidino-2-phenylindole (DAPI), blue) in sham and SNI DRG. Scale bar = 100 μm. **m, o** Representative example of high-magnification merge (×63), a puncta (yellow) can be seen in macrophages (F4/80[+] red cells), scale bar = 10 μm. Data are means ± S.E.M., n = 3 mice/group. ***P < 0.001, one-way ANOVA, post hoc Bonferroni

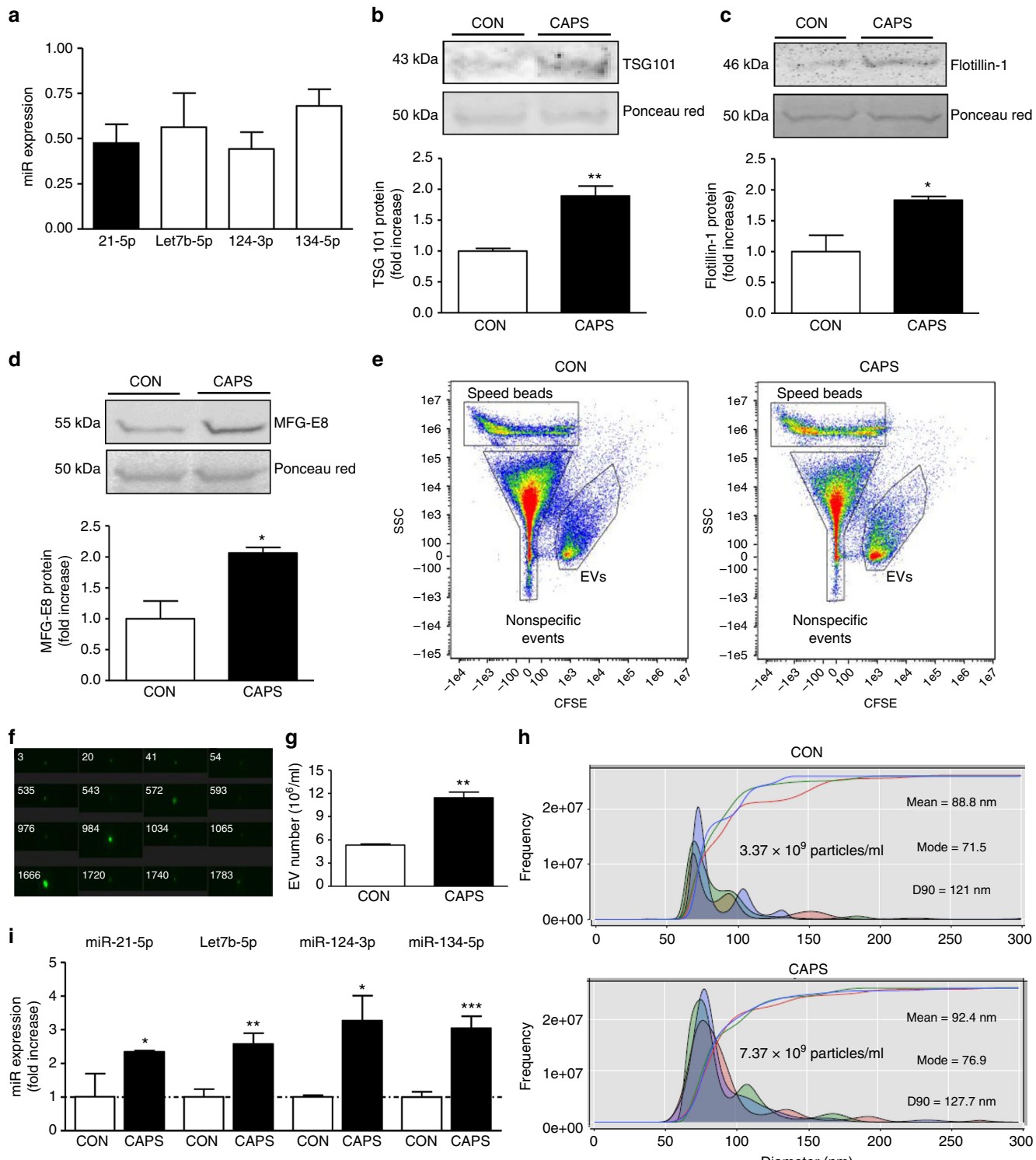

**Fig. 2** Capsaicin induces release of extracellular vesicles that include exosomes containing miRs from sensory neurons in culture. **a** Intracellular expression of miRs 21-5p, Let7b-5p, 124-3p, and 134-5p normalized to SNORD 202 (housekeeping non-coding RNA; n = 5 cultures). **b–d** Representative western blot and quantification of exosomal markers TSG101, Flotillin-1, and MFG-E8 in the culture media of DRG neurons incubated with buffer control (0.001% dimethyl sulphoxide (DMSO) in HEPES buffer + glucose 1 mg/ml; CON) or Capsaicin (1 μM; CAPS) for 3 h. Data are means ± S.E.M., n = 4 cultures; *P < 0.05 and **P < 0.01 compared to control, Student's t-test. **e–g** ImageStream™ analyses. **e** Pseudocolour dot plots of carboxyfluorescein succinimidyl ester (CFSE) fluorescence against side scatter (SSC) for EVs isolated from culture media of neurons incubated with buffer control or CAPS for 3 h. **f, g** Representative images and quantification of EVs. Data are means ± S.E.M., n = 4 cultures; **P < 0.01 compared to control, Student's t-test. **h** NanoSight detection of exosomes isolated from culture media of neurons incubated with buffer control or CAPS for 3 h. Representative outcome is shown. Each colored line represents a single frame recording with a total of 3 frames for each sample. The inset data indicate mean diameter, mode diameter and D90 values represent the diameter of 90% of the particles. **i** Expression of miRs 21-5p, Let7b-5p, 124-3p and 134-5p in the exosomal fraction of DRG neurons media treated with buffer control or CAPS for 3 h. Data are means ± S.E.M., n = 4 cultures; *P < 0.05, **P < 0.01 and ***P < 0.001 compared to control, Student's t test

sensitization[5–7]. Both central and peripheral sensitizations are fundamental for the generation of allodynia, hyperalgesia, and spontaneous pain[8].

The manipulation of neuron–macrophage/microglia communication is proving to be a viable instrument with which to halt the development of neuropathic pain, and both macrophage and microglia targets are being considered for novel therapeutic approaches[1, 4, 9].

Here we investigate the mechanisms by which neurons and macrophages communicate in the DRG and modify the inflammatory infiltrate after peripheral axon injury. Specifically, we focus our attention on the release of extracellular vesicles (EVs), including exosomes, from sensory neuron cell bodies in the DRG. Exosomes are EVs that are secreted by all types of cells, including immune cells and neurons[10]. While initially thought to be a cellular mechanism of waste disposal, EVs are now also considered to be highly specified enablers of intracellular and intercellular communication[11]. Exosomes derive from multivesicular bodies (MVBs) and secretory exosomes contain a specific cargo composition[10]. Current evidence indicates that MVBs are present in the cell bodies of sensory neurons in the DRG rather than in peripheral or central axonal terminals[12], suggesting that cell bodies may release EVs, including exosomes under appropriate conditions. However, evidence for the ability of primary sensory neurons to secrete exosomes is yet to be established. Although electrical excitability of the cell bodies in the DRG is not necessary for signal conduction to the central nervous system, their cell membranes are electrically excitable and peripherally generated spikes, which propagate centrally, invade, and provoke activity of the soma, which also has the capacity to fire spontaneously[13]. Recent in vivo imaging studies demonstrate that neuronal coupling in DRG contributes to pain hypersensitivity after peripheral injury[14].

Exosome cargo includes a variety of microRNAs (miRs), and recent evidence indicates significant dysregulation of miRs in the DRG and spinal cord after nerve injury[15–17]. These miRs can modulate nociception and, for instance, intrathecal delivery of miR-124, miR-103, and miR-23b attenuates inflammatory and neuropathic pain by altering intracellular neuronal, astrocytic, and microglial functions[18–20]. Conversely, miR-let7b exerts a pro-nociceptive effect via mediation of neuron–neuron cross-excitation. Following its activity-induced release by DRG neurons, miR-let7b activates TRPA1 channels, thus providing positive feedback for sensory neurons[21]. In addition, miR-134, which is also expressed in the DRG, is pro-nociceptive in chronic pain models[22] and miR-183 cluster controls neuropathic pain-regulated genes in DRG[23].

To date, however, much of our understanding regarding miR-mediated effects on pain mechanisms is based on the use of "unpackaged" miRs. In order to advance our knowledge, it is now critical to assess the effect of miRs in a biologically relevant setting in which they are present as exosomal cargo. In this study we assess whether sensory neuron cell bodies in the DRG secrete exosome-containing miRs as a means to communicate with infiltrated macrophages after peripheral nerve injury.

## Results

**Exosomes containing miRs are released from DRG neurons.** Several miRs are dysregulated in sensory neurons after spared nerve injury (SNI) and in particular the expression of miR-21 increases after sciatic nerve axotomy[24]. We confirmed that peripheral nerve injury induces upregulation of miR-21 expression in the lumbar DRG. Specifically, we observed relatively low expression of miR-21 in the cell bodies of sensory neurons under sham conditions (Figs. 1a, c, e) as well as contralateral to nerve

injury (Figs. 1d, e) and significant upregulation of miR-21 in ipsilateral sensory neurons 7 days after SNI (Figs. 1b, e). No miR-21 expression in DRG was detected with a control scrambled probe (Supplementary Figs. 1a–d). Expression of miR-21 was elevated 7 days after SNI in both large-diameter neurons (NF200+; Figs. 1f–h) and small-diameter peptidergic neurons (CGRP+; Figs. 1i–k) compared to either contralateral or ipsilateral sham neurons. Predictably, while very few F4/80+ cells were found in sham injury DRG (Figs. 1l, m), in the DRG ipsilateral to injury, infiltrating macrophages (F4/80+ cells) were observed in the vicinity of sensory neuron cell bodies containing miR-21 (Fig. 1n). In addition, some macrophages expressed miR-21 fluorescence in SNI DRG (Fig. 1o) and miR-21-positive macrophages were more abundant in DRG ipsilateral compared to contralateral SNI and ipsilateral sham injury (Supplementary Fig. 1e).

In order to confirm that the endogenous expression of miR-21 and other selected miRs could be detected in DRG, we measured intracellular expression in dissociated DRG cultures. Expression of miR-21-5p was comparable to miR-let7b, miR-124, and miR-134, which were selected as positive-control miRs (Fig. 2a). As we were interested in determining whether sensory neurons release EVs and miRs from their cell bodies following noxious-like activation, we treated cultured DRG with capsaicin. Incubation of capsaicin for 25 min and 3 h resulted in a significant accumulation, in the exosomal fraction of the culture media, of the endosomal sorting complex required for transport (ESCRT-I) component Tumor Susceptibility Gene 101 (TSG101; Fig. 2b and Supplementary Fig. 2a), which is an intracellular protein central for exosomal sorting from MVBs[25]. Capsaicin treatment for 3 h, but not 25 min, also induced a significant extracellular increase of the exosomal marker Flotillin-1 (Fig. 2c and Supplementary Fig. 2b), which belongs to the family of lipid raft-associated proteins and is involved in endosome and exosome recycling[26]. Similarly, capsaicin incubation for 3 h, but not 25 min, significantly increased the extracellular expression of the exosomal protein Milk Fat Globule-E 8 protein (MFG-E8; Fig. 2d and Supplementary Fig. 2a), which is an adhesion molecule and acts as an opsonin for cells to be engulfed by phagocytosis following binding to αvβ5 integrin in macrophages[27, 28]. Capsaicin incubation was not associated with neurotoxicity as determined by LDH assay (data not shown). Consistent with an activity-dependent release of exosomal markers, incubation of cultured DRG with depolarizing agent potassium chloride at 25 and 50 mM for 3 h promoted release of TSG101 and MFG-E8 (Supplementary Fig. 3). However, the highest concentration was required for Flotillin-1 band to become visible (Supplementary Fig. 3). Furthermore, treatment of pure sensory neurons in culture with capsaicin induced significant release of EVs compared to buffer incubation alone (Figs. 2e–g). We could readily observe neuronal EVs by ImageStream™ flow cytometry[29] (Figs. 2e, f) which were significantly elevated in capsaicin compared to control media (Fig. 2g). NanoSight tracking analysis confirmed that capsaicin incubation doubled the number of EVs in the media and revealed that both control and capsaicin media were rich in particles, 90% of which have a diameter below 125 nm with a mode of 74.7 nm and hence can be genuinely considered as exosomes (Fig. 2h and Supplementary Fig. 4a).

Expression analysis of miRs in the exosome fraction of cultured DRG media indicated that capsaicin significantly increased levels of miR-21-5p, let7b, miR-124, and miR-134 compared to control conditions (Fig. 2i). In keeping with this increase of miRs in EV fractions, in DRG whole-cell lysates, capsaicin induced a significant increase in *Dicer* mRNA and Dicer protein fragments isolated by immunoprecipitation (~90 and 66 kDa; Supplementary Figs. 4b, c). As the RNase type III enzyme Dicer is the rate-

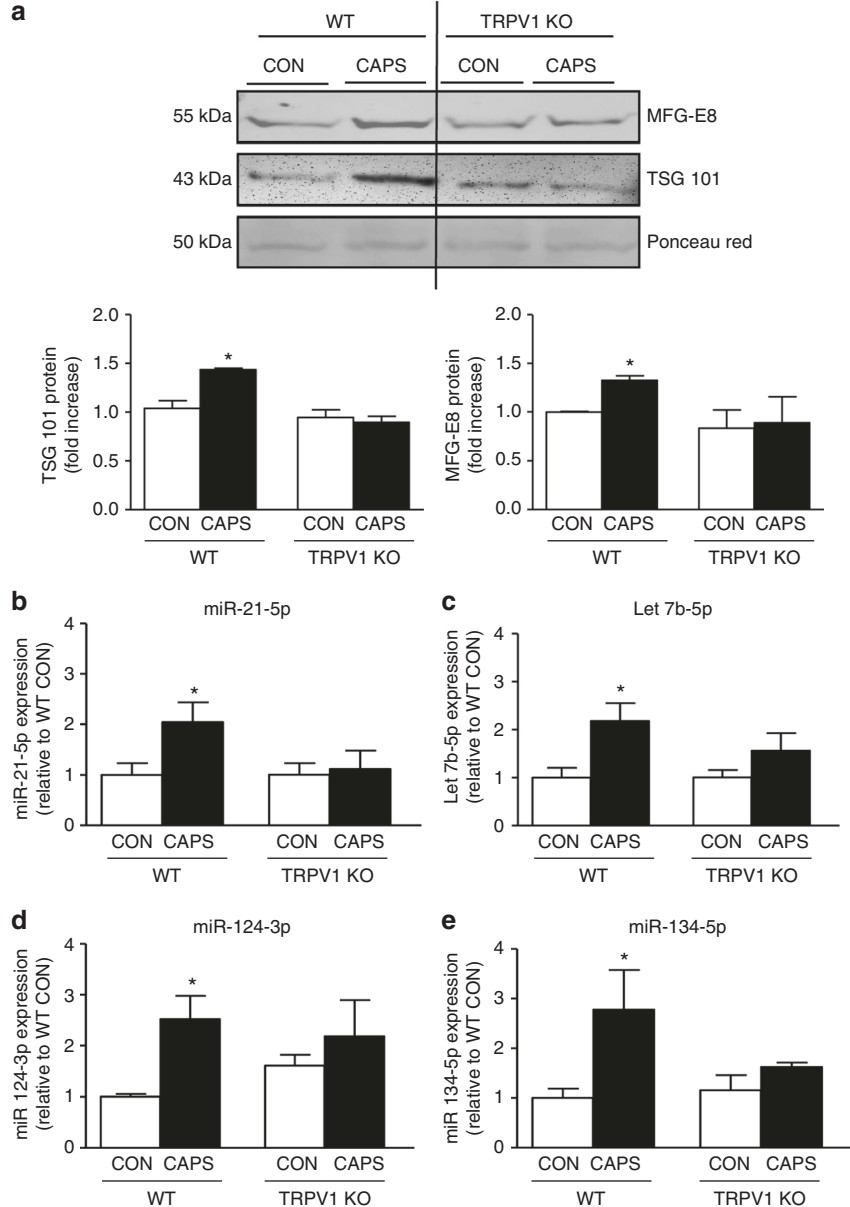

**Fig. 3** Exosomes release from sensory neurons is mediated by TRPV1 activation. **a** Representative western blot image and quantification of exosomal markers TSG101 and MFG-E8, in the media of cultured DRG neurons obtained from WT and TRPV1 KO mice and incubated with buffer control or CAPS for 3 h. **b–e** Expression of miRs in exosomal fraction of DRG media obtained from WT and TRPV1 KO mice. Data are means ± S.E.M., $n = 4$. *$P < 0.05$ compared to control WT, one-way ANOVA, post hoc Bonferroni test

limiting enzyme in the miR formation from pre-miR[30], these data suggest that capsaicin may trigger Dicer activity in sensory neurons. The effect of capsaicin was TRPV1 receptor-mediated. In fact, capsaicin-induced release of TSG101 and MFG-E8 positive EVs was significantly reduced when dissociated DRG obtained from mice deficient in TRPV1 were used (Fig. 3a). Similarly, the release of miR-21-5p, let7b, miR-124 and miR-134 measured in the exosomal fraction of wild-type (WT) DRG was absent in DRG obtained from TRPV1 knockout (KO) mice (Figs. 3b–e). As TRPV1 is exclusively expressed by neurons and not satellite cells[31], we can confidently attribute these differences in release between WT and TRPV1 KO mice as neuronal.

These in vitro data indicate that exosomes containing miRs, such as miR-21-5p, are released from the cell bodies of TRPV1-expressing sensory neurons. In the in vivo scenario of the DRG after peripheral nerve injury, nociceptive neuron-derived exosomes may interact with macrophages, which infiltrate in response to nerve injury. Such macrophages may phagocytose miR-21-containing exosomes, especially those expressing the adhesion molecule MFG-E8, which we have identified in our EV preparations.

**Sensory neuron-derived EVs are phagocytosed by macrophages.** To assess whether macrophages phagocytosed neuron-derived exosomes, resulting in functional transfer of miR-21-5p, we incubated peritoneal macrophages with exosome-enriched media derived from pure sensory neurons in culture and made the following three observations. Firstly, EVs were effectively transferred into macrophages as confirmed by ImageStream™ analysis using fluorescently labeled vesicles. Neuron-derived EVs (CFSE-labeled; green) were rapidly engulfed by primary macrophages (F4/80+ in Fig. 4a) as demonstrated by both augmented positive events in the dot plots and representative images

(Fig. 4b). Quantification at single-cell level revealed that macrophages accumulated significantly more EVs following incubation with capsaicin-derived media compared to control media (Fig. 4c). Secondly, EV phagocytosis had an impact on macrophage phenotype, as evident by an increase in mRNA for inducible nitric oxide synthase (iNOS, Nos2, M1 marker) and decrease in mRNA for CD206 (mannose receptor, Mrc1, M2 marker) in macrophages incubated with neuron-derived EVs

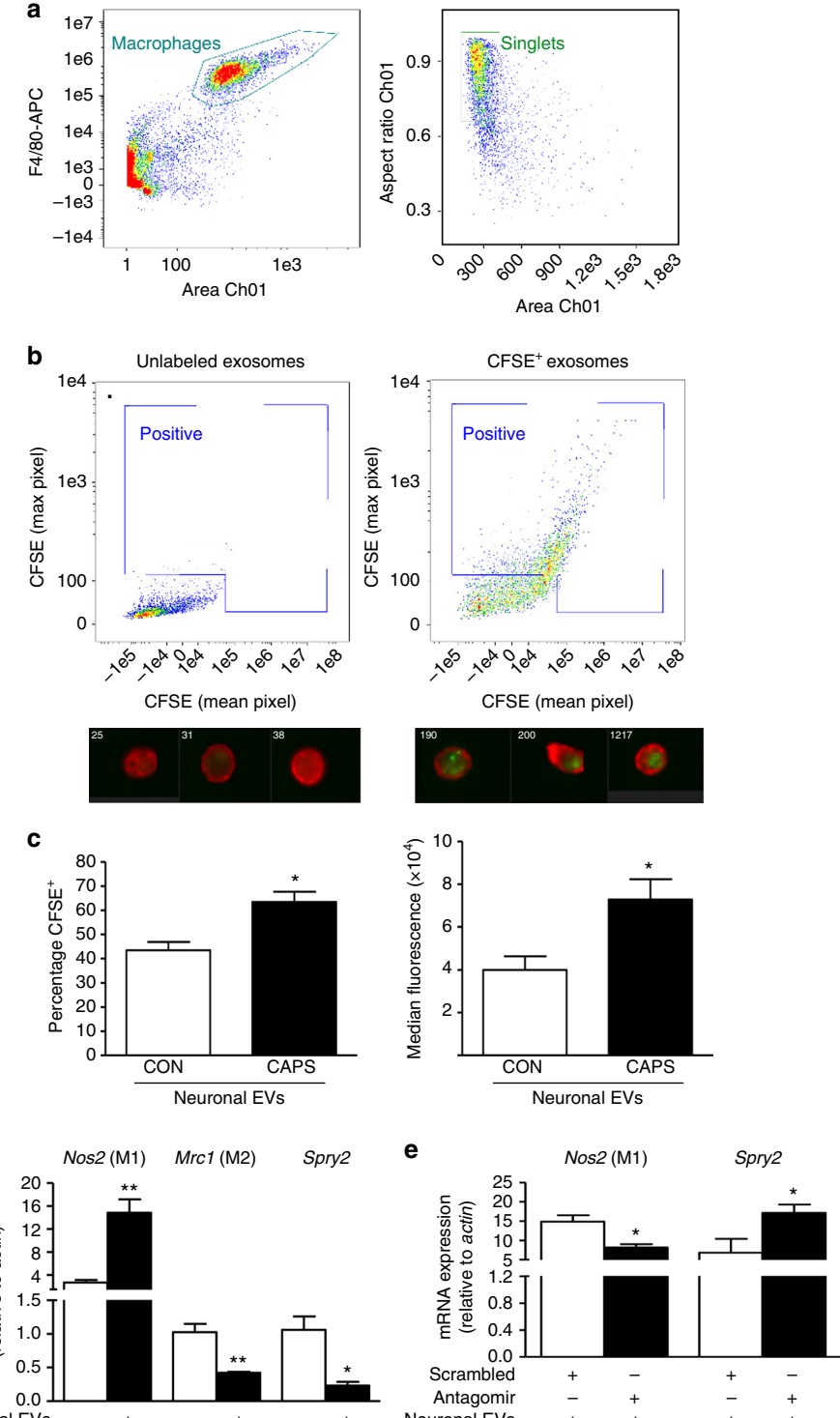

**Fig. 4** Exosomes released from pure sensory neurons after capsaicin are phagocytosed by macrophages. **a**, **b** Representative scatterplots (gating strategy) and ImageStream™ images showing EVs (CFSE-labeled, green) uptake by macrophages (F4/80+, red). EVs were isolated from culture media after incubation of pure DRG neurons with buffer control or CAPS for 3 h. **c** Percentage-positive and median fluorescence intensity of CFSE+ macrophages incubated with neuron-derived EVs. Data are means ± S.E.M., n = 4. *P < 0.05, compared to control, Student's t-test. **d** Nos2, Mrc1, and Spry2 mRNA expression levels in macrophages incubated with and without exosomes derived from CAPS-treated pure DRG neurons. **e** Nos2 and Spry2 mRNA expression levels in macrophages transfected with miR-21-5p antagomir or scrambled oligomer and incubated with EVs derived from CAPS-treated pure DRG neurons. Data are means ± S.E.M., n = 3; *P < 0.05 and **P < 0.01 Student's t-test

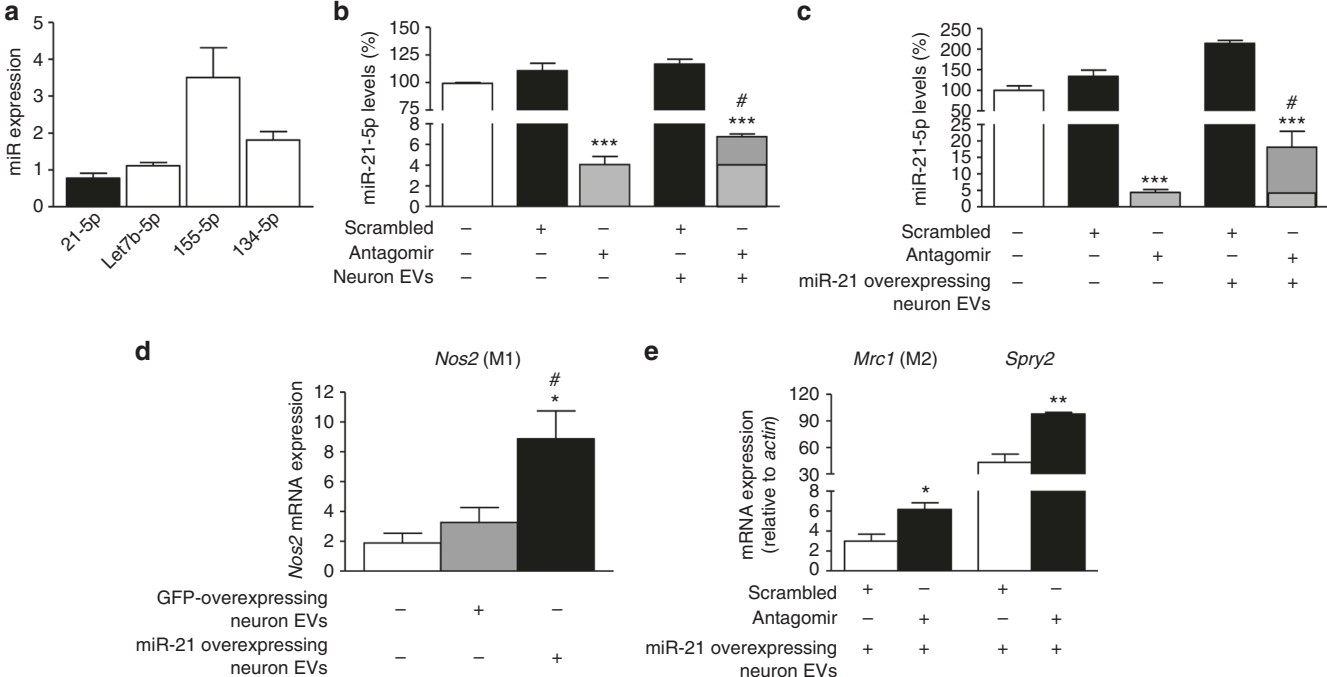

**Fig. 5** Expression of miR-21-5p is increased in antagomir-transfected macrophages exposed to capsaicin-released exosomes. **a** Peritoneal macrophage intracellular expression of miRs 21-5p, Let7b-5p, 155-5p, and 134-5p normalized to SNORD 202 (housekeeping non-coding RNA; $n = 4$ cultures). **b** miR-21-5p expression in macrophages transfected with miR-21-5p antagomir or scrambled oligomer and incubated with or without EVs derived from CAPS-treated pure DRG neurons. **c** miR-21-5p expression in macrophages transfected with miR-21-5p antagomir or scrambled oligomer and incubated with or without miR-21-overexpressing EVs derived from CAPS-treated DRG neurons. Data are means ± S.E.M., $n = 3$; ***$P < 0.001$, compared to scrambled oligomer; #$P < 0.05$, compared to macrophages not exposed to EVs, one-way ANOVA, post hoc Bonferroni test. **d** Expression of Nos2 mRNA in macrophages incubated with GFP-overexpressing or miR-21-overexpressing neuronal EVs. Data are means ± S.E.M., $n = 3$; *$P < 0.05$ compared to macrophages not exposed to EVs; #$P < 0.05$ compared to GFP-derived EVs, one-way ANOVA, post hoc Bonferroni test. **e** Mrc1 and Spry2 mRNA expression levels in macrophages transfected with miR-21-5p antagomir or scrambled oligomer and incubated with miR-21-overexpressing EVs derived from CAPS-treated DRG neurons. Data are means ± S.E.M., $n = 3$; *$P < 0.05$ and **$P < 0.01$ Student's *t*-test

compared to macrophages not exposed to exosomes (Fig. 4d). Consistent with possible miR-21 transfer in macrophages, the expression of Sprouty2 mRNA (Spry2) a known miR-21 target[24] was downregulated (Fig. 4d). Finally, inhibition of miR-21-5p prevented the effect of neuron-derived exosomes on cell phenotype as treatment of macrophages with a specific miR-21-5p antagomir resulted in lower expression of Nos2 concomitant to higher expression of Spry2, compared to treatment with scrambled oligomer (Fig. 4e). We have observed that macrophages expressed basal constitutive levels of miR-21-5p, although to a much lesser extent than other miRs, such as miR-155-5p (Fig. 5a). Thus, to further investigate possible transfer of miR-21, we silenced miR-21-5p (Figs. 5b, c) and incubated macrophages with exosomes isolated from either non-viral transfected or miR-21-overexpressing sensory neurons[24]. Under both conditions, we observed that in antagomir-transfected macrophages incubation of neuron-derived EVs produced a significant increase in miR-21-5p expression (Figs. 5b, c). Macrophage incubation with miR-21-overexpressing neuron EVs was associated with upregulation of Nos2 (Fig. 5d), whereas in the presence of the antagomir, incubation of miR-21-overexpressing neuron EVs resulted in higher expression of Mrc1 and Spry2 (Fig. 5e).

Altogether, these in vitro observations indicate that sensory neurons can transfer miR-containing exosomes to macrophages and this transfer results in changes of cell phenotype as well as intracellular levels of miR-21-5p and known miR-21 gene targets.

**Overexpression of miR-21-5p promotes M1 macrophage phenotype**. To define the effect of increased intracellular miR-21-5p

on macrophage polarization, we transfected primary peritoneal macrophages with either fluorescence-labeled miR-21-5p (miR-21-5p mimic) or the scrambled sequence termed N4. Macrophage transfection with miR-21-5p mimic produced more than a 90% yield efficiency in F4/80+ cells and resulted in a significant increase in miR-21-5p expression relative to controls (Figs. 6a, b). Transfection with N4 displayed the same efficacy as miR-21-5p mimic, but, as expected, did not result in increased expression of miR-21-5p (Fig. 6b). The expression of Spry2 was downregulated in miR-21-5p mimic-transfected macrophages compared to control transfection and N4-transfected cells (Fig. 6c).

Subsequent analysis of polarization markers in miR-21-5p-transfected macrophages revealed upregulation of several pro-inflammatory markers compared to N4 transfection and control transfection. Specifically, we detected an increase of both protein and mRNA for iNOS and transcription nuclear factor κB subunit p65 (NF-κB p65, Rela; Figs. 6d, e). Furthermore, miR-21-5p transfection, more significantly than N4 and control transfection, reduced the transcriptional levels of Mrc1 and Arginase-1 (Arg1; Fig. 6e). Consistent with these intracellular changes, extracellular levels of pro-inflammatory cytokines, tumor necrosis factor (TNF)-α and interleukin (IL)-6, were higher in the media obtained from miR-21-5p-transfected macrophages compared to either control transfection or N4 media (Fig. 6f).

These in vitro transfection data support evidence for a pro-inflammatory role of miR-21[32] by showing that an increase in intracellular miR-21-5p induces macrophage polarization toward a pro-inflammatory M1 phenotype.

In order to substantiate our finding, we performed flow cytometry analyses of miR-21-5p-transfected macrophages and

observed a significant shift in favor of the M1 population. Specifically, macrophages that were transfected with miR-21-5p were significantly polarized toward M1 (CD206⁻CD11c⁺) compared to both N4-transfected cells and control transfection (Figs. 7a, e) while the proportion of M2 cells (CD206⁺ CD11c⁻) was not significantly altered (Figs. 7a, f). Notably, the numbers of CD45⁺, F4/80⁺ CD11b⁺, and CD206⁺ CD11c⁺ cell populations were comparable after control, N4 and miR-21-5p transfection (Figs. 7a–d).

**Inflammatory macrophages infiltrate ipsilateral DRG.** One week after nerve injury, inflammatory macrophages infiltrate in the DRG in higher numbers than in sham-injured DRG[9]. By performing flow cytometry analysis of leukocytes isolated from Day-7 sham-operated and neuropathic mice, we demonstrated a significant infiltration of leukocytes (CD45⁺ cells) in DRG ipsilateral to peripheral nerve injury compared to contralateral DRG (Supplementary Figs. 5b, c). This infiltration was also apparent under sham conditions, but to a lesser extent (Supplementary

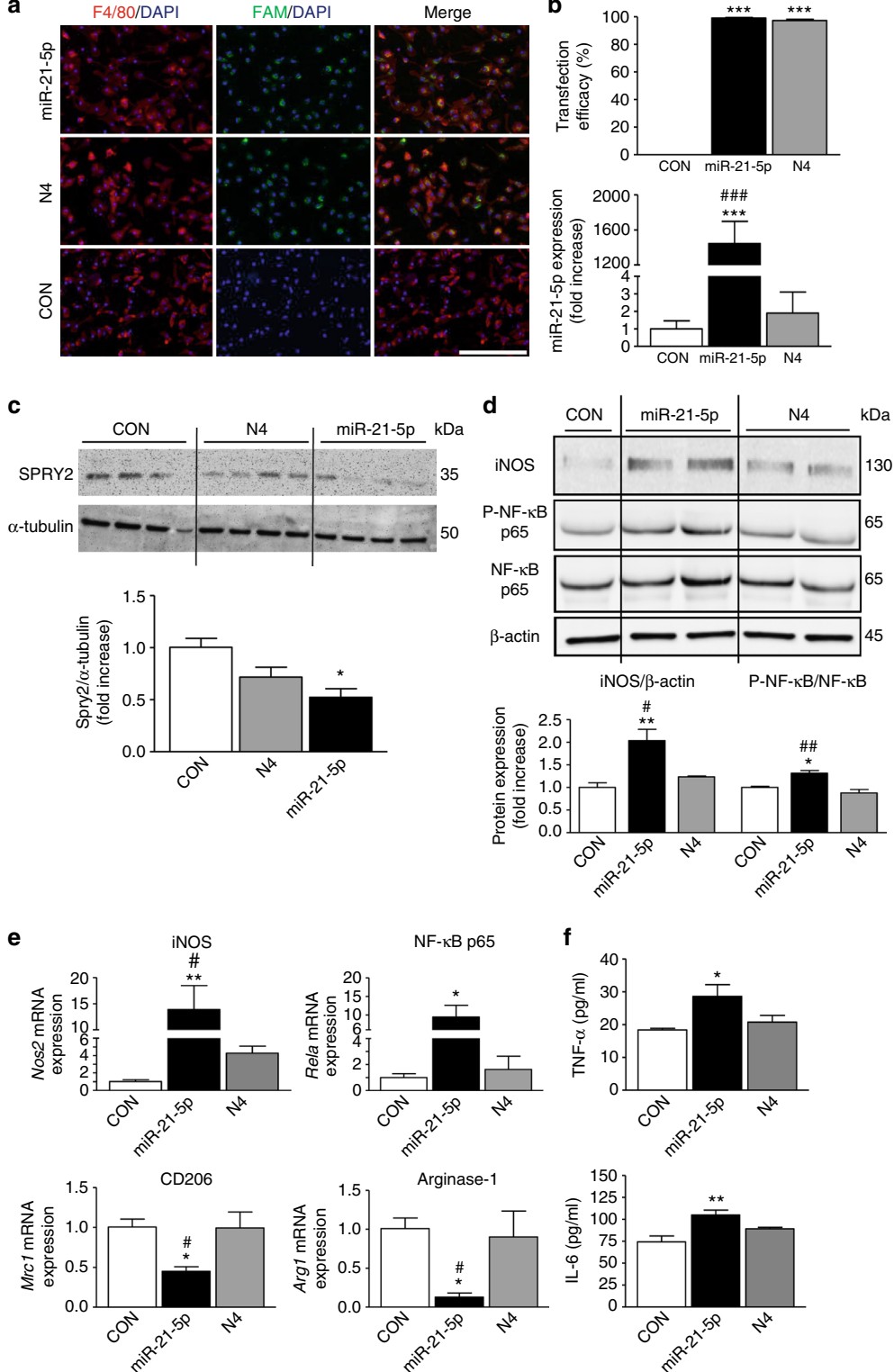

Figs. 5a, c). Specifically, F4/80$^+$CD11b$^+$ macrophages, and especially the CD206$^+$CD11c$^+$ subset, were present in higher numbers in ipsilateral compared to contralateral DRG in both sham and injured conditions (Supplementary Figs. 5a, b). The ipsilateral vs. contralateral difference in macrophage numbers was even more significant in injured DRG (Supplementary Figs. 5a, b, d). In particular, CD206$^-$CD11c$^+$ cells (M1 macrophages) were more numerous in the ipsilateral injured DRG than DRG contralateral to injury and sham DRG (Supplementary Fig. 5e), while CD206$^+$CD11c$^-$ cells (M2 macrophages) were significantly less abundant in injured compared to sham ipsilateral DRG (Supplementary Fig. 5f). SNI is therefore associated with infiltration of pro-inflammatory macrophages in the DRG ipsilateral to injury, which is less pronounced in DRG contralateral to injury or under sham conditions.

**miR-21 antagomir prevents nociceptive hypersensitivity.** Having observed that (i) macrophages that had infiltrated into the DRG after injury displayed a pro-inflammatory M1 phenotype; (ii) DRG sensory neurons were able to release exosomes containing miR-21-5p in an activity-dependent manner; (iii) miR-21-5p overexpression in macrophages was associated with M1 phenotype, we hypothesized that in vivo the ongoing nociceptive neuron activity contributes to M1 polarization of macrophages through exosome release, which serves as a neuron–macrophage communication mediator.

Specifically, we postulated that neuronal miR-21 could contribute to the nociceptive hypersensitivity and influence the nature of the inflammatory infiltrate in the DRG after peripheral nerve injury. In order to test this hypothesis, we took a dual approach by (i) performing prolonged intrathecal delivery of the miR-21-5p antagomir and (ii) generating sensory neuron conditional miR-21 null mice. We observed that intrathecal delivery of the miR-21-5p antagomir, but neither the scrambled oligomer nor transfecting agent (vehicle), significantly prevented the development of nerve injury-associated nociceptive hypersensitivity from day 2 to 7 by ~50% (Fig. 8a). Critically, delivery of the miR-21-5p antagomir did not have an effect on mechanical thresholds contralateral to injury (Fig. 8b). The administration of the miR-21-5p antagomir to naive mice significantly enhanced expression of Spry2 compared to the scrambled oligomer (Supplementary Fig. 6a) and reduced miR-21-5p expression in the lumbar DRG (Supplementary Fig. 6b), confirming efficient delivery of the construct. As both the miR-21-5p antagomir and scrambled oligomer were fluorescently tagged, we examined their distribution. No fluorescence was observed in the spinal cord (Supplementary Fig. 7). However, both compounds had reached the ipsilateral and contralateral DRG where they accumulated preferentially in the cell bodies of sensory neurons (Figs. 8c, d, i and Supplementary Fig. 8). The constructs were found in up to 20% of the macrophages (F4/80$^+$ cells, Figs. 8e, f, i) and 5% of satellite cells (GFAP$^+$; Figs. 8g–i). Furthermore, immunohistochemical analysis revealed that miR-21-5p antagomir treatment

significantly reduced the number of macrophages (F4/80$^+$ expressing cells) in DRG ipsilateral to injury compared to treatment with the scrambled oligomer (Figs. 8j, k). Systemic administration of the same dose of miR-21-5p antagomir for 7 days via subcutaneous pumps altered neither SNI mechanical hypersensitivity nor the number of macrophages compared to scrambled oligomer (Supplementary Fig. 9), suggesting that the intrathecal oligomer was effective locally.

**miR-21 antagomir modulates macrophage phenotype in DRG.** We then went on to assess the phenotype of macrophages present in the DRG following peripheral nerve injury under the intrathecal treatment conditions. Flow cytometry analysis of the leukocyte population (CD45$^+$ cells) in the DRG revealed, following injury, a significant higher number of M1 macrophages in ipsilateral compared to contralateral DRG when the scrambled oligomer was delivered continually for 7 days (Figs. 9a, c–f), similar to that observed in untreated neuropathic DRG (Supplementary Fig. 5). However, delivery of the miR-21-5p antagomir altered the macrophage profile in DRG ipsilateral to injury. We detected a reduction in CD45$^+$ cells (leukocytes; Fig. 9b) and no longer observed a significant difference in this population between ipsilateral and contralateral DRG (Fig. 9c). We also observed a significant reduction in F4/80$^+$CD11b$^+$ cells (macrophages) as well as CD206$^-$CD11c$^+$ cells (M1 phenotype) compared to ipsilateral DRG treated with the scrambled oligomer (Figs. 9a, b, d, e). M2 macrophages (CD206$^+$CD11c$^-$) were significantly reduced in ipsilateral compared to contralateral DRG following injury when mice were treated with the scrambled oligomer, but not the miR-21-5p antagomir (Fig. 9f). Thus, 7-day-intrathecal administration of a miR-21-5p antagomir, which accumulated predominantly in the cell bodies of sensory neurons in the DRG, resulted in a reduction in pro-inflammatory macrophage infiltration in the DRG and a significant behavioral antinociceptive effect.

**miR-21 deletion in sensory neuron prevents hypersensitivity.** In order to establish the cellular source of miR-21 and further substantiate the proposed mechanism of neuroimmune interaction, we silenced miR-21 expression selectively in sensory neurons, and achieved a significant 50% downregulation of miR-21 expression in the miR-21 conditional KO (cKO) compared to WT littermates DRG (Fig. 10a). After peripheral nerve injury, in cKO we observed an attenuation of ipsilateral nociceptive hypersensitivity compared to WT, which reached significance on days 5–7 (Fig. 10b). Contralateral thresholds were comparable between cKO and WT (Fig. 10b). In the lumbar DRG of cKO the number of CD45$^+$ cells, which had infiltrated by day 7 from nerve injury was significantly lower than in WT (Fig. 10c). Macrophage numbers (F4/80$^+$CD11b$^+$ cells; Figs. 10d, e) and M1 cells (CD206$^-$CD11c$^+$; Fig. 10f) were significantly reduced in cKO compared to WT ipsilateral DRG. Notably, numbers of M2 macrophages (CD206$^+$CD11c$^-$) were elevated in cKO ipsilateral DRG

**Fig. 6** Transfection of peritoneal macrophages with miR-21-5p "mimic" induces upregulation of pro-inflammatory markers. **a** Peritoneal macrophage transfection for 48 h with FAM-labeled miR-21-5p mimic or scrambled control N4, and co-localization of miR-21-5p and N4 (FAM-labeled, green) with F4/80 (macrophage marker, red); nuclear stain DAPI (blue). Objective ×20, scale bar = 200 μm. **b** Transfection efficacy expressed as percentage of cells double-positive for F4/80 and FAM-labeled miR-21-5p mimic or N4; miR-21-5p expression in transfected macrophages revealed by qPCR. Data are means ± S.E.M., n = 6 coverslips. ***$P < 0.001$ compared to CON; ###$P < 0.001$ compared to N4, one-way ANOVA, post hoc Bonferroni. **c** Overexpression of miR-21-5p in macrophages induces a significant reduction of Spry2 protein revealed by western blot. **d** Representative western blot and quantification of iNOS and P-NF-κB protein levels induced by overexpression of miR-21-5p in macrophages. Data are means ± S.E.M., n = 4 for each group. *$P < 0.05$ and **$P < 0.01$ compared to CON; #$P < 0.05$ and ##$P < 0.01$ compared to N4, one-way ANOVA, post hoc Bonferroni. **e** mRNA expression levels for pro-inflammatory (Nos2, Rela) and anti-inflammatory (Mrc1, Arg1) mediators following macrophage transfection with miR-21-5p mimic and N4. Data are means ± S.E.M., n = 6 for each group. *$P < 0.05$ and **$P < 0.01$ compared to CON; #$P < 0.05$, compared to N4, one-way ANOVA, post hoc Bonferroni. **f** TNF-α and IL-6 levels in media of transfected macrophages. Data are means ± S.E.M., n = 6; *$P < 0.05$ and **$P < 0.01$ compared to CON, one-way ANOVA, post hoc Bonferroni

compared to WT DRG (Fig. 10g). To identify possible gene targets that are regulated by neuron-derived miR-21 to drive the macrophage response, we performed a genome-wide microarray

analysis of macrophages (F4/80$^+$CD11b$^+$ cells) isolated from L4 and L5 DRG of WT and cKO at 7 days after SNI (Supplementary Fig. 10a). We found that 2207 genes were statistically and

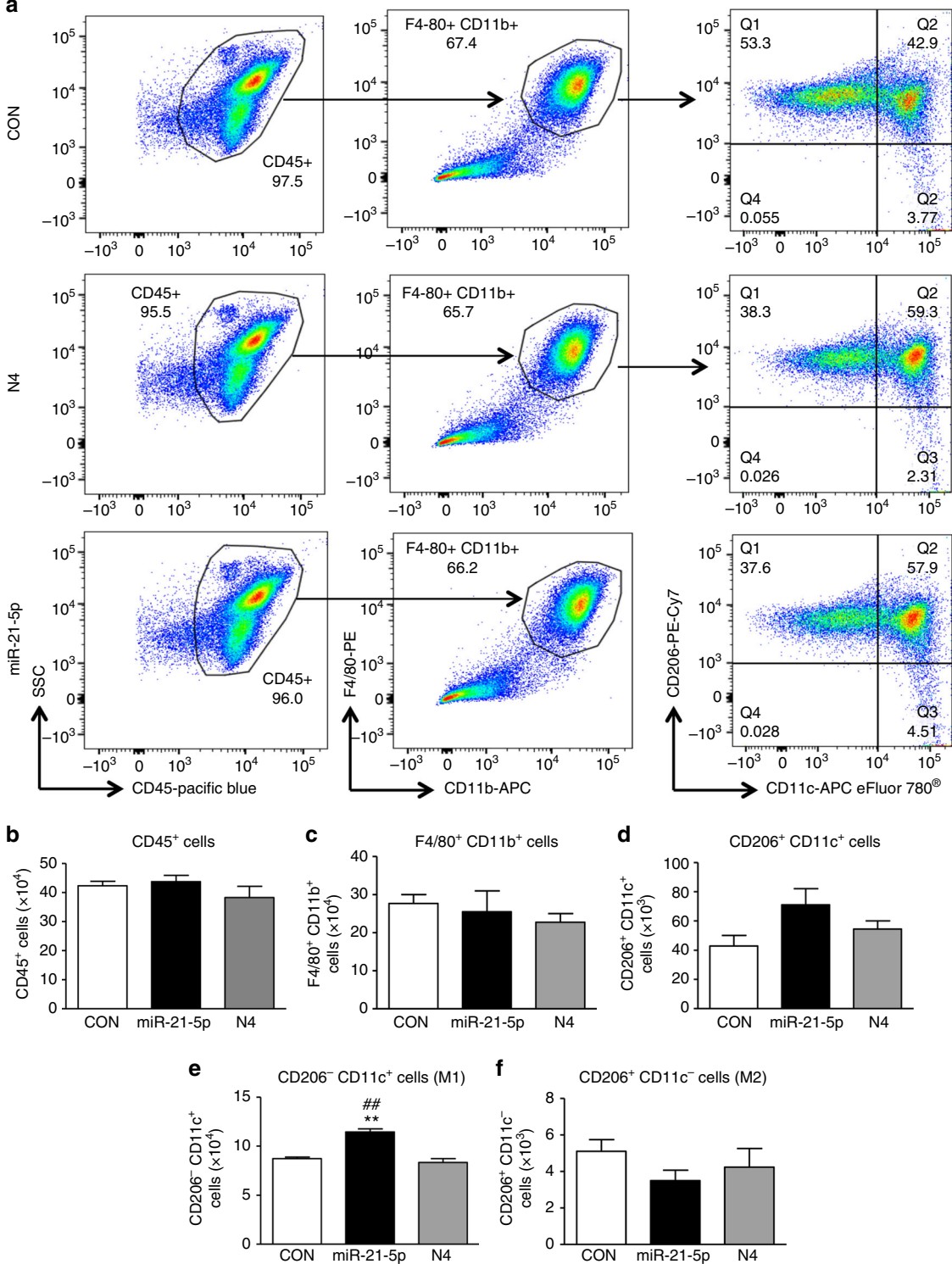

**Fig. 7** Transfection of peritoneal macrophages with miR-21-5p mimic reduces M2 and favors M1 phenotype. **a** Representative scatterplots of peritoneal macrophages transfected with N4 or miR-21-5p mimc for 48 h. Cells were immunolabeled with antibody–fluorophore conjugates against CD45, CD11b, F4/80, CD206, and CD11c. Macrophages were defined as CD45$^+$CD11b$^+$F4/80$^+$ cells, and their fluorescence intensity for CD206 and CD11c labels were used to define M1 (CD206$^-$CD11c$^+$) and M2 (CD206$^+$CD11c$^-$) phenotypes. Numbers in gates refer to percentage of positive cells for each specific marker. **b–f** Bar charts represent numbers of leukocytes (CD45$^+$ cells), macrophages (CD11b$^+$F4/80$^+$) and (CD206$^+$CD11c$^+$), M1 macrophages (CD206$^-$CD11c$^+$) and M2 macrophages (CD206$^+$CD11c$^-$). Data are means ± S.E.M., $n = 4$ for each group. **$P < 0.01$ compared to CON; ##$P < 0.01$ compared to N4, one-way ANOVA, post hoc Bonferroni

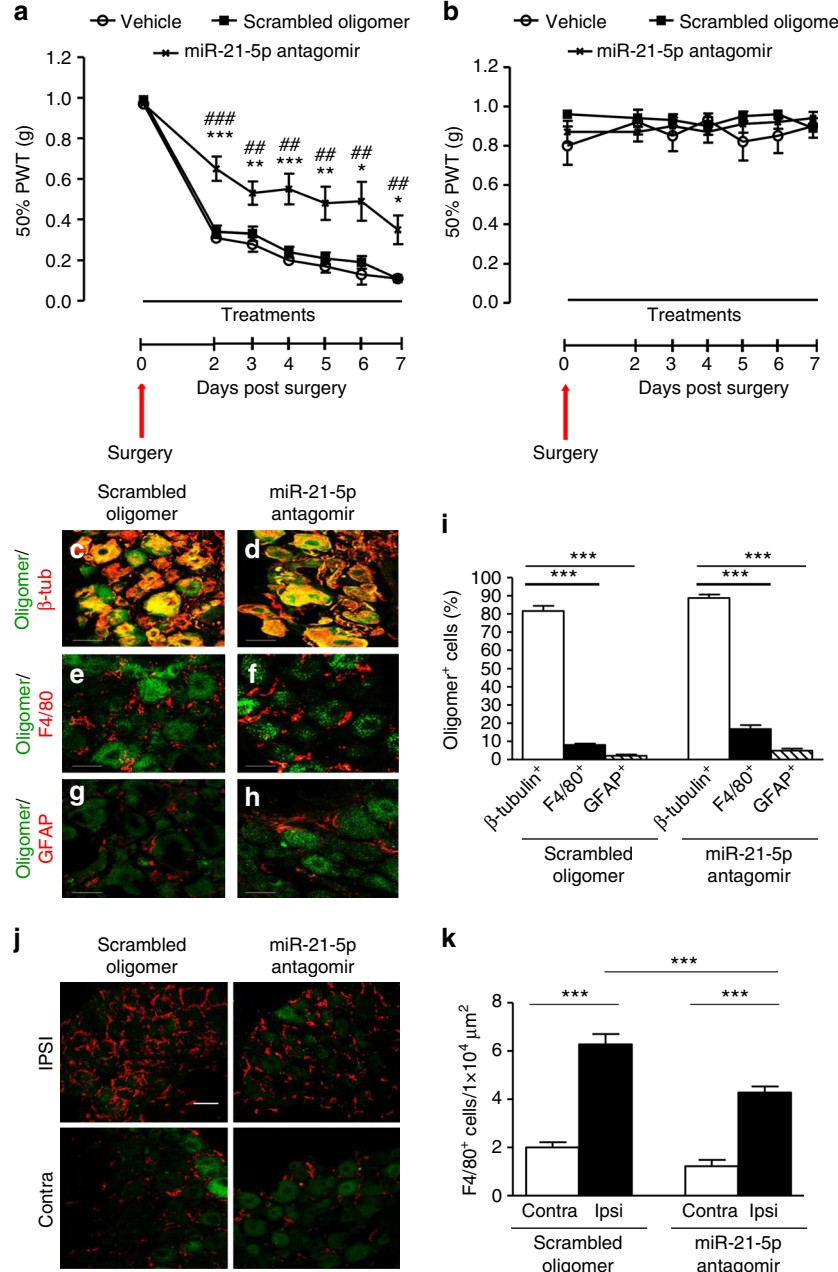

**Fig. 8** Intrathecal delivery of miR-21 antagomir prevents mechanical hypersensitivity and reduces macrophage numbers in the DRG following SNI injury. **a** Effect of continuous intrathecal delivery of the miR-21-5p antagomir (12 pmol/day) for 7 days on the development of mechanical hypersensitivity. **b** No effect on contralateral nociceptive thresholds of miR-21-5p, scrambled oligomers, or vehicle. Data are presented as 50% of paw withdrawal thresholds (PWT); means ± S.E.M., $n = 6$ mice in vehicle group and 12 mice in oligomer-treated groups. $*P < 0.05$, $**P < 0.01$, $***P < 0.001$ compared to vehicle; $^{##}P < 0.01$, $^{###}P < 0.001$ compared to scrambled oligomer, two-way ANOVA followed by Tukey test. Scrambled oligomer and miR-21-5p antagomir (green) distribution in L5 DRG and co-localization with the neuronal marker β-tubulin (red; **c**, **d**, **i**) macrophage marker F4/80 (red; **e**, **f**, **i**), or the satellite cell marker GFAP (red; **g**–**i**). Scale bar = 50 μm. **j** Macrophage infiltration (F4/80, red) in ipsilateral and contralateral L5 DRG. Scale bar = 50 μm. **k** Quantification of F4/80⁺ cells in L4 and L5 DRG ipsilateral and contralateral to injury following intrathecal delivery of either the scrambled oligomer or miR-21-5p antagomir. Data are means ± S.E.M., $n = 4$ for each group. $***P < 0.001$, one-way ANOVA, post hoc Bonferroni

significantly regulated in the sorted macrophage population (Supplementary Fig. 10b) and a number of pathways were significantly perturbed by neuronal miR-21 absence (Supplementary Fig. 10c). To gain further insight from these pathways, we examined the cohort of regulated genes and found that three known target genes of miR-21 were significantly regulated in the macrophages (Supplementary Fig. 10d). Specifically, we have identified genes coding for two binding proteins *Acta2, Tpm1,*

and one transcription factor, *znf288*, that are responsible for cytoskeleton remodeling and cell survival (Supplementary Fig. 10d).

## Discussion

This study provides novel evidence for the dysregulation of non-coding miRs in sensory neurons after peripheral axon injury,

which regulates the nature of the inflammatory infiltrate in the DRG microenvironment and exerts a significant impact on the development of neuropathic hypersensitivity. Specifically, peripheral nerve injury induced upregulation of miR-21 in ipsilateral DRG neurons, which was associated with ipsilateral mechanical hypersensitivity. The intrathecal delivery of a miR-21-5p

antagomir resulted in (i) downregulation of miR-21-5p expression and upregulation of Spry2 in DRG; (ii) prevention of the development of ipsilateral mechanical hypersensitivity; and (iii) reduction of inflammatory macrophage number in DRG. As predicted, systemic delivery of the same dose of the antagomir did not alter neuropathic hypersensitivity, suggesting that intrathecal

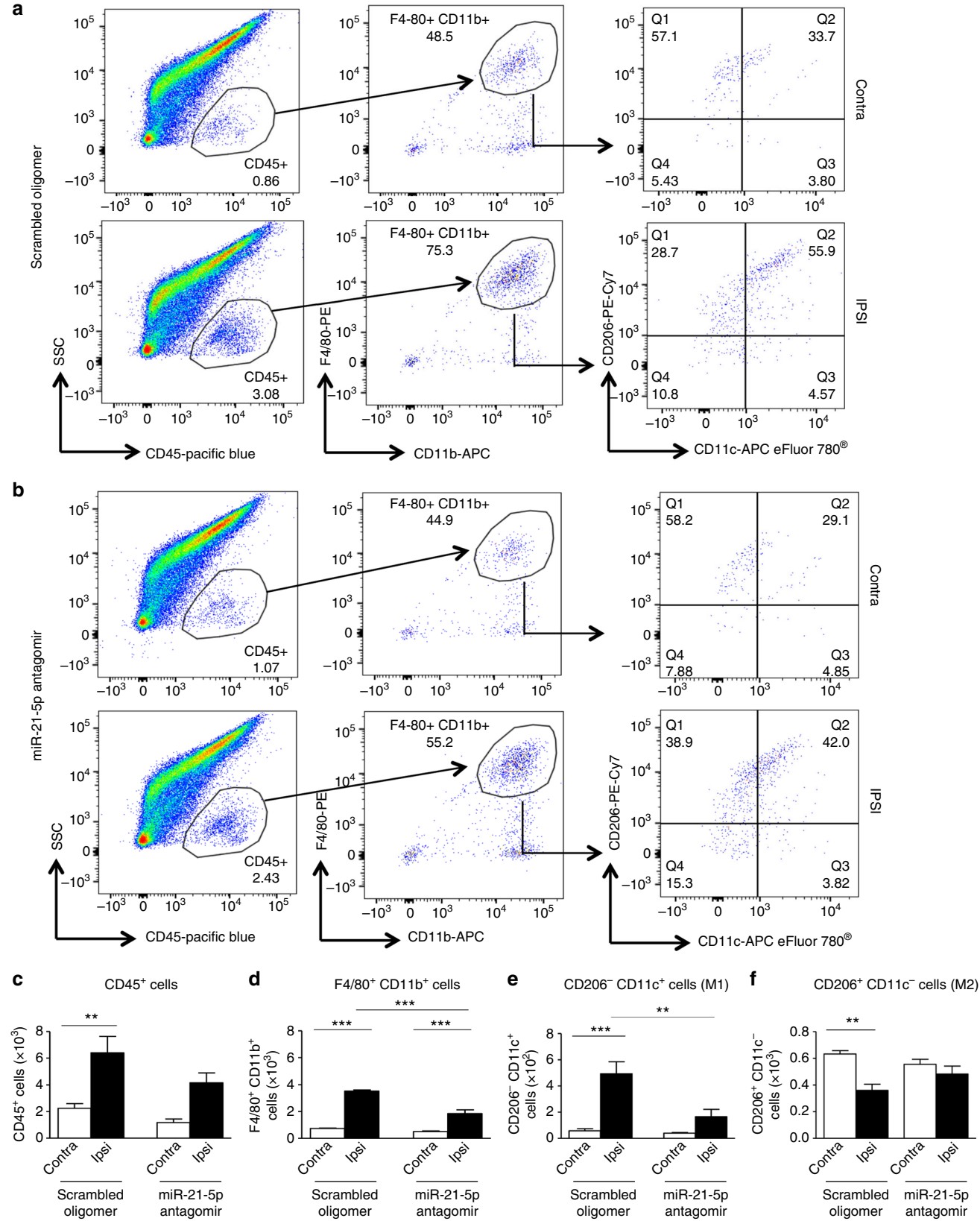

delivery of the antagomir acted locally and systemic distribution of the same dose of antagomir possibly diluted the active concentration of the oligomer. It would be worthwhile to test whether higher systemic doses of the antagomir display antihyperalgesic efficacy and target miR-21 at sites additional to the DRG.

Consistent with the intrathecal antagomir data, deletion of miR-21 expression in sensory neurons resulted in development of a less severe mechanical hypersensitivity and a marked reduction of inflammatory macrophage infiltration in the DRG. Moreover, selective deletion of neuronal miR-21 was associated with a significant presence of anti-inflammatory macrophages that displayed significant alteration in known miR-21-5p target genes.

Furthermore, we show in vitro that capsaicin stimulation of sensory neurons causes the release of exosomes containing miR-21-5p, that when phagocytosed by macrophages, promotes an increase in the expression of pro-inflammatory genes and proteins. Therefore, we conclude that the release of exosomes containing miRs is a plausible mechanism of neuron–macrophage communication in the DRG after nerve trauma. Whether EVs transfer pre-miR-21 and/or induce transcription of miR-21 in macrophages remain interesting possibilities. However, these data are indicative of neuronal miR-21 being a critical regulator, either directly or indirectly, of large number of cellular processes that underpin the macrophage phenotypes.

The cell bodies of sensory neurons in the DRG are invaded by action potentials traveling from the periphery to the spinal cord. Following peripheral nerve injury, ectopic impulses generated in the DRG become substantial and neuron-to-neuron coupling contributes to neuropathic pain[13, 14, 33]. Sensory neurons can modulate immune cells' function, for instance, by releasing inflammatory neuropeptides that mediate neurogenic inflammation[34]. We provide in vitro evidence that nociceptive neuron activity results in the secretion of EVs, including exosomes, which in vivo could serve the novel function of providing a mode of communication with nearby macrophages. The formation of MVBs and secretion of exosomes remain exclusive to sensory neuron cell bodies, as peripheral nervous system axons are unlikely to anteroretrogradely transport MVBs, which are rarely detected in PNS axons in vivo[12]. In dissociated DRG culture, neurons are mixed with satellite cells; however, we provide imaging evidence that pure sensory neurons release EVs, including exosomes, after capsaicin, which activates neuronal TRPV1 receptors leading to upregulation of intracellular Dicer. Thus, we demonstrate that sensory neuron cell bodies can perform regulated exocytosis of EVs in response to noxious-like activation and specific intracellular signaling pathways. Furthermore, detection of exosomal markers TSG101, Flotillin-1, and the adhesion molecule MFG-E8 in dissociated DRG culture media indicates the presence of exosomes, which can interact with phagocytic cells such as macrophages. Exosomal cargo can vary depending on cell type and stimulus, and we observed that sensory neuron-derived exosomes contain several miRs including miR-21, which is upregulated in the DRG following peripheral nerve injury. miR-21 has been shown to play a significant intracellular role in promoting neurite outgrowth through downregulation of the Spry2 protein[24] and it is conceivable that miR-21 regulates targets

in sensory neurons that can influence nociceptors' function directly or indirectly at peripheral and central terminals.

Our data provide evidence for an extracellular role of neuron-derived miR-21, which is released as part of exosomal cargo. Released upon DRG neuron soma activity in exosomes, when phagocytosed and overexpressed by macrophages, miR-21 would polarize macrophages toward a pro-inflammatory over an anti-inflammatory phenotype as evident by several validated markers. We acknowledge evidence that miR-21 may upregulate IL-10 in macrophages after 24 h-TLR4 stimulation, which mimics an infective status[35]. It is conceivable that the modulatory roles of miR-21 on macrophage polarization are plastic as well as dependent on time, stimulus, and specific microenvironments. Following traumatic nerve injury we could demonstrate a functional link between overexpression/deletion of miR-21 with pro-/anti-inflammatory macrophages both in vitro and in vivo.

Inflammatory macrophages infiltrate the DRG after peripheral axon injury and release cyto/chemokines that contribute to neuronal sensitization[3, 9]. We suggest that in vivo neuron–macrophage transfer of exosomes containing miR-21 may serve as a regulator of macrophage phenotype and consequentially promotes a pro-nociceptive environment. Herein, we make two observations that demonstrate that DRG neurons are a critical cellular source of exosomes containing miR-21. Specifically, stimulation of sensory neurons in vitro with capsaicin induced release of miR-21-5p in a TRPV1-mediated manner, suggesting that nociceptive neurons are a likely source of miR-21. Intrathecally delivered miR-21-5p antagomir in vivo, which significantly prevented the development of neuropathic hypersensitivity and showed preferential tropism toward neuronal cells over satellite cells and macrophages. Notably, the antagomir showed limited distribution to the spinal cord and probably higher centers.

Consistent with a neuronal origin of miR-21, the deletion of miR-21 expression in sensory neurons was associated with anti-nociceptive behavior, polarization of macrophages toward an anti-inflammatory phenotype, and alteration in macrophages of known miR-21-5p target genes that are responsible for cytoskeleton remodeling and cell survival. As the macrophage phenotype is associated with change in their shape that depends on contractility within the actin cytoskeleton[36], future studies will determine potential direct association between cytoskeletal changes and downstream effects of miR-21-5p on macrophage polarization and function.

In summary, this study demonstrates not only a novel mechanism by which neuron–macrophage communication occurs, but also a potential function of DRG neuron excitability in the context of nerve trauma-associated pain. Cell bodies of sensory neurons in the DRG release exosomes and selected miRs, including miR-21-5p, following specific activation of nociceptive neurons by capsaicin. Once released, exosomes can be phagocytosed by infiltrating macrophages where elevated miR-21-5p expression is accompanied by an increase in pro-inflammatory and decrease in anti-inflammatory phenotype.

These data have multiple implications and can open a series of opportunities. Firstly, we suggest that targeting this mode of

**Fig. 9** Intrathecal delivery of miR-21-5p antagomir reduces immune cell recruitment and number of M1 macrophages in DRG following SNI injury. Representative scatterplots of immune cells sorted from pools of contralateral and ipsilateral L4 and L5 DRG obtained from SNI mice intrathecally treated with scrambled oligomer (**a**) or miR-21-5p antagomir (**b**) as in Fig. 8. Cells were gated on CD45$^+$, F4/80$^+$, and CD11b$^+$. Macrophages were defined as CD11b$^+$F4/80$^+$ and further analyzed for the M2 (CD206$^+$CD11c$^-$) and M1 (CD206$^-$CD11c$^+$) phenotypes as indicated by arrows. Numbers in gates refer to the percentage of positive cells for each specific marker. Bar charts represent absolute number in DRG of leukocytes (CD45$^+$; **c**), macrophages (CD11b$^+$F4/80$^+$; **d**), M1 macrophages (CD206$^-$ CD11c$^+$; **e**), and M2 macrophages (CD206$^+$ CD11c$^-$; **f**). Statistical analysis was performed on data obtained from four independent experiments. Data are means ± S.E.M., $n = 4$ for each group. **$P < 0.01$ and ***$P < 0.001$, one-way ANOVA, post hoc Bonferroni

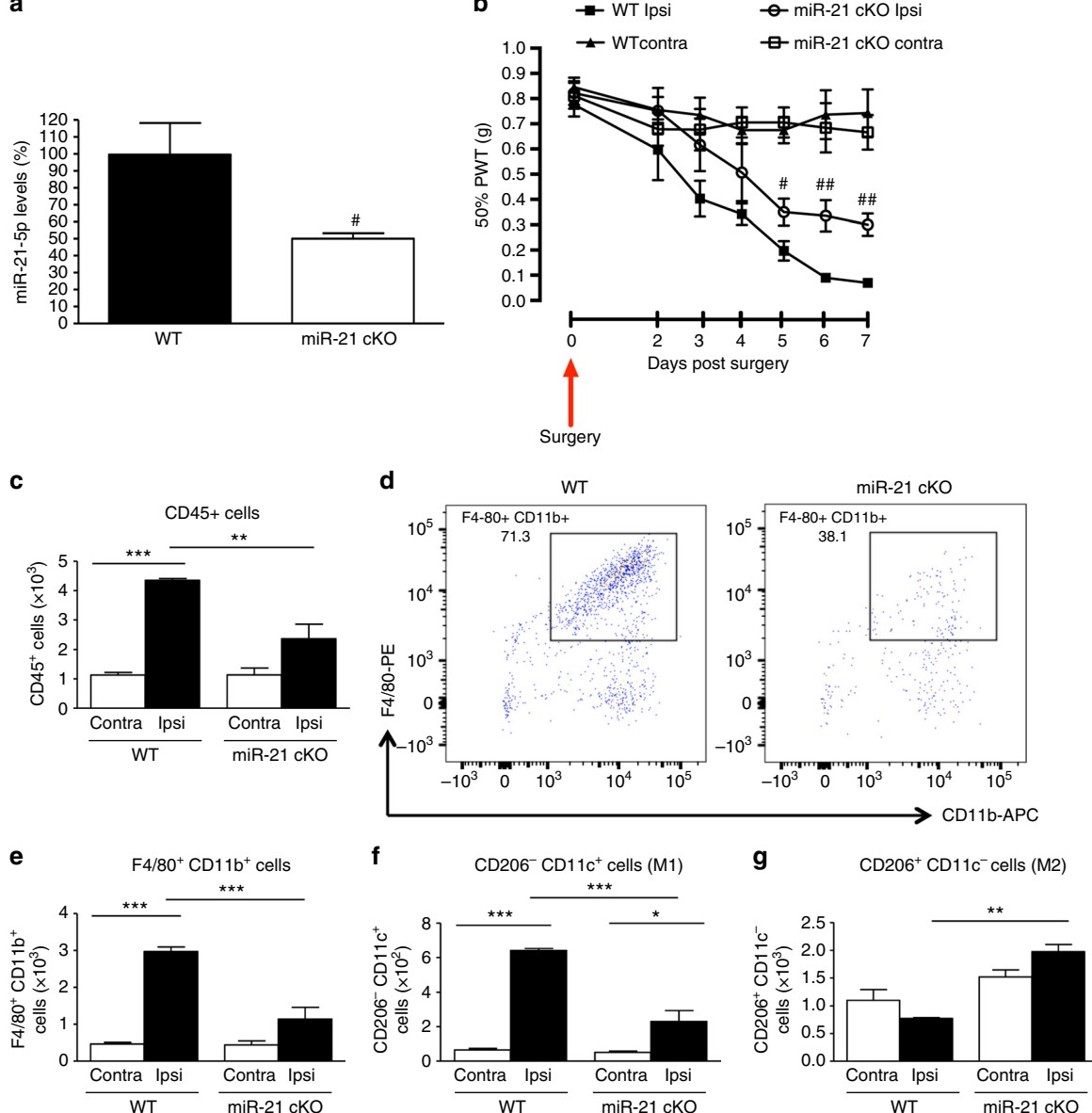

**Fig. 10** Conditional deletion of miR-21 in sensory neurons prevents the development of mechanical hypersensitivity and is associated with polarization of macrophages toward an anti-inflammatory phenotype. **a** Downregulation of miR-21-5p levels in DRG of miR-21 cKO compared to WT littermate mice. Data are expressed as means ± S.E.M., $n = 5$ mice for each group. [#]$P < 0.05$ compared to WT, Student's $t$-test. **b** Effect of miR-21 deletion in DRG sensory neurons on the development of mechanical hypersensitivity following SNI. Data are presented as 50% of paw withdrawal thresholds (PWT); means ± S.E.M., $n = 10$ mice. [#]$P < 0.05$, [##]$P < 0.01$, compared to WT ipsilateral, two-way ANOVA followed by Tukey test. **c** Bar charts represent absolute number of leukocyte in DRG. **d** Representative scatterplots of immune cells sorted from pools of ipsilateral L4 and L5 DRG obtained from WT mice or miR-21 cKO mice, on day-7 post SNI injury. Numbers in gates refer to the percentage of positive cells for each specific marker. **e** Number of macrophages, **f** M1 macrophages (**g**) and M2 macrophages. Statistical analysis was performed on data from two independent experiments. Data are expressed as means ± S.E.M., $n = 4$ for each group. *$P < 0.05$, **$P < 0.01$, and ***$P < 0.001$, one-way ANOVA, post hoc Bonferroni

neuron–macrophage communication could prove to be a promising and innovative analgesic strategy and provide an alternative to current treatments of pain following nerve trauma, which show limited efficacy at present. Secondly, specific elements of the exosome cargo could be targeted and we provide here an example of such approach centered on miR-21-5p. Finally, the possibility of delivering exosomes in a tissue-specific manner may spare essential neuron–macrophage communication, providing a potential therapy with limited side effects. We note that preparations of exosomes derived from dendritic cells have entered clinical trials for immunotherapy in cancer patients and both production and characterization of clinical-grade

exosome products are being actively pursued[11]. The application of this technology could be translated to the treatment of neuropathic pain.

## Methods

**Animals**. Experiments were carried out in 8–12-week-old male C57BL/6 mice according to the United Kingdom Animals (Scientific Procedures) Act 1986, and following the guidelines of the Committee for Research and Ethical Issues of the International Association for the Study of Pain. Adult TRPV1 KO mice and WT control littermates were kindly donated by Stuart Bevan, King's College London. Animals were housed with ad libitum access to food and water and maintained on a 12 h light/dark cycle. Experimental study groups were randomized and blinded.

**Generation of sensory neuron-specific miR-21 null mice.** The mouse line miR-21, with a KO first conditional allele, was sourced from the Jackson Laboratory (accession number 36060). Mice were bred with FLP-Deleter mice and subsequently with Avil-Cre driver mice for conditional ablation in DRG[37]. The genotyping strategy of the line has been described previously[37, 38]. Conditional KO and WT littermates (8–10-week-old males) were used. Procedures were performed according to the Guide for the Care and Use of Laboratory Animals of the National Institutes of Health. Mice were maintained at the EMBL Mouse Biology Unit, Monterotondo, Italy, in accordance with Italian legislation (Art. 9, 27 January 1992, no 116) under licence from the Italian Ministry of Health.

**Behavioral testing.** Static mechanical withdrawal thresholds were assessed by application of calibrated von Frey monofilaments (0.02–1.0 g) to the hind paw plantar surface. Testing started with the application of a 0.07 g filament and each paw was assessed alternately between application of increasing stimulus intensity until a withdrawal response was achieved or application of 1.0 g filament failed to induce a response, in order to avoid tissue damage. The 50% paw withdrawal threshold (PWT) was determined by increasing or decreasing stimulus intensity, and evaluated using Dixon's "up–down" method. Experiments were performed blind.

**Induction of neuropathy.** Mice received a SNI[39] under isofluorane anesthesia. Briefly, the skin and muscle of the left thigh were incised to expose the sciatic nerve and its three terminal branches. The common peroneal and tibial nerves were tightly ligated and the distal nerve stump was removed, while the sural nerve was left intact. In sham-injured mice, sciatic nerve was exposed but not ligated or excised. PWTs were examined prior to and daily from days 2 to 7 after surgery.

**Intrathecal and subcutaneous delivery of oligomers.** Intrathecal cannula catheterization was performed on the same day of SNI surgery. Under anesthesia a small laminectomy was made over the thoracic spinal cord[40]. A polyethylene catheter (Alzet, Charles River Ltd, UK) was inserted under the dura mater in the lumbar enlargement and attached to a subcutaneous osmotic pump (Alzet 1007D, Charles River Ltd). For subcutaneous oligomer administration pumps were not connected to the cannula. An LNA-based miR-21-5p inhibitor and scrambled control oligomer were custom-made as fluorescein amidite (FAM)-labeled compounds by Exiqon (Denmark). Sequences are reported in Supplementary Table 1. The oligomers were mixed (1:5 v/v) with i-Fect™ in vivo transfection reagent (Neuromics, 2B Scientific, UK) and delivered for 7 days at 12 pmol/day. At the end of treatments, catheter and pump were checked to ascertain efficient delivery.

**In situ hybridization and immunohistochemistry.** Sections (10 μm) were obtained from fresh frozen L4 and L5 DRG using a cryostat (Leica CM3050, UK) and mounted onto Superfrost Plus slides (Thermo Scientific, UK). Cryosections were fixed in 4% paraformaldehyde (PFA) and treated with proteinase K (5 μg/ml; Sigma, UK). The sections were acetylated with 1.35% triethanolamine/0.25% acetic anhydride/0.18% HCl and pre-hybridized in 50% formamide, 4× saline sodium citrate, 0.5 mg/ml yeast tRNA, 1× Denhardt's solution prior to hybridization with a digoxigenin (DIG)-labeled probe complementary to mouse miR-21 (0.5 pmol, LNA miRCURY probe; Exiqon). Scrambled probes were used as controls. Probe sequences: miR-21: TCAACATCAGTCTGATAAGCTA; scrambled: GTGTAA-CACGTCTATACGCCCA. Following hybridization, sections were incubated with a mouse anti-DIG horseradish peroxidase antibody (1:500, Abcam, UK). The in situ hybridization signals were enhanced using a tyramide amplification system labeled with Alexa-488 (Invitrogen, UK). For subsequent immunohistochemistry (IHC) sections were incubated with the following primary antibodies: rabbit anti-mouse NF200 (1:1000, Abcam), mouse anti-mouse CGRP (1:500, Enzo Life Sciences), rat anti-mouse F4/80 (1:200, AbD Serotec), and secondary antibody (1:1000, Invitrogen Alexa-568 goat anti-rat for F4/80 and Alexa-568 goat anti-mouse for NF200 and CGRP), and were coverslipped using Vectashield mounting medium with DAPI (Vector Labs, UK).

For IHC in perfuse-fixed tissues, mice were transcardially perfused with saline solution, followed by PFA in 0.1 M phosphate buffer and lumbar spinal cord and L4 and L5 DRG were excised. Transverse sections of spinal cord (20 μm) and DRG (10 μm) were taken using a cryostat (Bright Instruments) and thaw-mounted onto SuperFrost Plus microscope slides (VWR). Sections were incubated with a rat anti-mouse F4/80 (1:200, AbD Serotec), rabbit anti-β-III tubulin (1:1000, Abcam), mouse anti-GFAP (1:400, Millipore), followed by an anti-rabbit Alexa Fluor 568 secondary antibody (1:1000, Invitrogen). The immunoreactivity was visualized using a Zeiss LSM710 confocal microscope and images were acquired using the LSM software (Zeiss, UK). Positive cells were quantified in areas of $25 \times 10^4$ or $1 \times 10^4$ μm² areas using ImageJ software (version 1.46r, Wayne Rasband, National Institutes of Health, Bethesda, MD). At least four sections from three mice per group were used. For flow cytometry analysis, L4 and L5 DRG pooled from six sham-injured or SNI mice were collected in F-12 Nutrient Mixture (Ham; Gibco) supplemented with dispase (3 mg/ml; Roche), collagenase type IV (0.1%; Worthington), and DNAse (200 U/ml; Roche). DRG were triturated in DMEM Nutrient Mixture F-12 Ham (Sigma-Aldrich) containing heat inactivated fetal bovine serum 10% (HI-FBS; Gibco) and 1% penicillin/streptomycin (Gibco; DRG

medium). Cell suspensions were then centrifuged at 800 RPM for 5 min and pellets resuspended in fresh medium (1.5% BSA in HBSS, Gibco).

**Primary cultures of dissociated DRG neurons.** DRG were collected and placed into F-12 Nutrient Mixture (Ham; Gibco) supplemented with 0.1% collagenase type IV (Worthington). Thereafter, DRG were triturated and cell suspensions centrifuged at 600 RPM for 6 min. Pellets were resuspended in fresh DRG medium supplemented with NGF (10 ng/ml; Promega), and plated on poly-L-ornithine (100 μg/ml, Sigma-Aldrich)-pre-coated glass coverslips. Cultures (10,000–22,500 cells/well) were incubated at 37 °C for 24 h. In the experiments where miR-21 was overexpressed, cultured neurons were transduced with a green fluorescent protein (control) or miR-21 lentiviral vector[24] and were incubated at 37 °C for 72 h.

**Primary cultures of pure sensory neurons.** Non-neuronal cells were separated from neurons using anti-biotin microbeads and MACS technology (Miltenyi Biotech)[41]. Briefly, DRG were dissociated using 3 mg/ml dispase (Roche), 0.1% collagenase (Sigma-Aldrich) and 200 U/ml DNAseI (Roche) in F-12 medium (Life Technologies). Thereafter, tissues were homogenized and the supernatants filtered through a 70-μm filter and centrifuged for 8 min at 1000 RPM. Pellets were washed, resuspended, and incubated in MACS buffer containing 0.5% w/v BSA and non-neuronal biotin antibody cocktail for 5 min on ice. After wash, cells were incubated with anti-biotin microbeads (Miltenyi Biotech) and purified through LD columns in the QuadroMACS separator. The eluates were centrifuged at 1000 RPM for 8 min and pellets resuspended in DRG medium. Plates were coated with 1:10 polylysine followed by 1:10 GFR matrigel (BD Bioscience) in F-12. Approximately 8000 cells/well were incubated at 37 °C for 24 h.

On the experimental day, neurons were either incubated with 0.001% DMSO or capsaicin (1 μM) for 25 min and 3 h or KCl (25 and 50 mM) for 3 h. After stimulation, an aliquot of the supernatants was collected for LDH Cytotoxicity Assay (Pierce, Thermofisher) and the remaining volume centrifuged at 13,000 g at 4 °C for 2 min to remove apoptotic bodies and cell debris. Thereafter, exosomes were isolated by ultracentrifugation (100,000 g), while cells were collected for Dicer immunoprecipitation, western blot analysis, and real-time PCR.

**EV isolation and analysis.** Pure neuron culture media were centrifuged at 13,000 g for 2 min at 4 °C to remove apoptotic bodies and cell debris. Supernatants were incubated with CellTrace™ CFSE dye (1:1000, Invitrogen) for 10 min on ice. Then, samples were ultracentrifuged at 100,000 g at 4 °C for 1 h. EVs deposited at the bottom of each tube were subjected to ImageStream™ analysis or incubated with macrophages for uptake experiments. Acquisition on the ImageStream^x MKII was conducted at slow flow rate and ×60 magnification, with the "Remove Beads" option switched on. The 488 nm laser was set at 200 mW, the side scatter laser was set at 70 mW, and CFSE fluorescence was detected in channel 2. Data are expressed as vesicles/ml (after ensuring stabilization of flow rate). EV size distribution analysis was performed using an NS300 Nanoparticle Tracker with 488 nm scatter laser and high-sensitivity camera (Malvern Instruments Ltd., Malvern, UK). For each sample, particle scatter was recorded three times for 60 s each under flow conditions (arbitrary speed 50) at camera level 16 and analysis threshold 5, using the NTA 3.2 acquisition and analysis software.

**Primary macrophage in culture.** Macrophages were obtained by lavage of the peritoneal cavity with 1% penicillin/streptomycin sterile saline, plated, and allowed to adhere. Thereafter, non-adherent cells were removed by washing and adherent macrophages covered with red phenol-free complete DMEM (Gibco) supplemented with 10% HI-FBS, 1% pen/strep, and 1% sodium pyruvate.

For mimic transfection, cells (500,000/well) were transfected with FAM-labeled miR-21-5p mimic or scrambled control N4 (500 ng; Exiqon), using a Lipofectamine® 3000 Transfection Reagent (Invitrogen), as an alternative to a DNA plasmid. Lipofectamine® 3000 Transfection Reagent was used as controls. Transfected cells were cultured for 48 h at 37 °C prior fixation with PFA for 20 min. After washing, coverslips were incubated with rat anti-mouse F4/80 antibody (1:200, AbD Serotec, UK) followed by Alexa Fluor 546 goat anti-rat (1:1000, Invitrogen), and were coverslipped in Vectashield mounting medium with DAPI. Six coverslips per group were analyzed using a fluorescent microscope (Zeiss), and transfection efficacy (%) after 48 h was calculated using the following formula for the correct transfection efficacy as described[42]:

$$\text{Transfection efficacy}(E) = [(E/D) \times 100]/X_D$$

where $D$ represents the number of cells used, $E$ is the number of F4/80⁺ cells expressing FAM-labeled miR-21-5p or N4, and $X_D$ is the proliferation coefficient, that is the ratio of the number of cells on the coverslips on analysis day to those counted on transfection day.

For antagomir transfection, cells ($1 \times 10^6$/well) were allowed to adhere for 6 h at 37 °C and transfected with either miR-21-5p antagomir or scrambled control oligomer (1 μg; Exiqon). Sequences of the oligomers are reported in Supplementary Table 1. Antagomir-transfected cells were cultured for 48 h at 37 °C and then exposed to EVs derived from either non-viral transfected or miR-21-overexpressing sensory neurons stimulated with capsaicin. Macrophages not incubated with EVs were used as controls. Culture media were then removed, while

cell lysates were obtained using a lysis/binding solution provided by mirVana miRNA Isolation Kit (Ambion). Small RNAs were isolated and miR levels detected by quantitative polymerase chain reaction (qPCR).

For flow cytometry analysis, plated cells were detached with $Ca^{2+}/Mg^{2+}$ free phosphate buffer saline containing EDTA (10 mM), centrifuged at 2,000 RPM for 8 min at 4 °C, and pellets were then resuspended in 1.5% BSA in 1× HBSS. TNF-α and IL-6 levels (pg/ml) were quantified in the culture media of macrophages using enzyme-linked immunosorbent assay (ELISA) kits (R&D Systems®).

**Imaging neuron-derived vesicles' incubation with macrophages**. Adherent peritoneal macrophages ($2 \times 10^6$ cells/well) were cultured for 24 h at 37° and then incubated with CFSE-stained EVs derived from pure DRG neurons for further 1 h. Cells were then washed and resuspended in buffer (10 mM EDTA in DPBS−/− supplemented with 3% FBS) containing anti-mouse F4/80-APC (1:200, eBioscience) for 30 min. After fixation (BD Bioscience fixation buffer) and centrifugation, cells were acquired on the ImageStream$^x$ MKII, at slow flow rate and ×60 magnification, with the 488 nm laser set at 200 mW, the 633 nm laser set at 100 mW, and the side scatter laser set at 3.7 mW. F4/80-APC fluorescence was detected on channel 11, EV CFSE fluorescence was detected on channel 2, brightfield images were acquired on channels 1 and 9, and side scatter was detected on channel 6. Appropriate single fluorophore controls were used for compensation, and macrophages treated with unlabeled vesicles were used as a fluorescence minus one gating control for CFSE positivity. Singlet macrophages were gated based on F4/80 expression and brightfield area and aspect ratio. Data are expressed as percentage of macrophages positive for CFSE and median fluorescence intensity of all singlet macrophages.

**Real-time PCR**. Intracellular miRNAs and Dicer mRNA levels in cultured DRG: Neurons were lysed with a mirVana miRNA Isolation Kit (Ambion). Total and small RNA-enriched fractions were isolated and RNA-eluted using RNase-free water. Both concentration and purity were estimated using a NanoDrop ND-100 Spectrometer (ThermoFisher Scientific). For miR detection, each small RNA template sample was diluted to 5 ng/μl using nuclease-free water, and cDNA synthesized using the miRCURY LNA™ Universal cDNA Synthesis kit II (Exiqon). PCR for miRs 21-5p, Let7b-5p, 124-3p, and 134-5p was performed using ExiLENT SYBR® Green master mix (Exiqon) in a LightCycler 480 (Roche). Primer sequences are reported in Supplementary Table 2. Duplicates of cycling thresholds (CTs) were averaged and the relative quantities of miRNAs calculated using the $2^{-\Delta\Delta CT}$ method and normalized to snoRNA202 (Assay ID 001232, TaqMan™ MicroRNA assays). For quantification of Dicer mRNA levels, PCR was performed using a LightCycler FastStart DNA MasterPlus SYBR Green I kit (Roche) in a LightCycler 480 (Roche). Primer sequences are reported in Supplementary Table 3. Duplicate CTs were averaged at the results analyzed by the $2^{-\Delta\Delta CT}$ method using Actb as a housekeeper gene.

Extracellular miRNAs in the exosomal fraction of cultured DRG: To measure miRNA release, DRG primary neurons (10,000/well) were incubated with 0.001% DMSO or capsaicin and exosomes were isolated from supernatants using miRCURY™ Exosome Isolation Kit (Exiqon) and small RNAs purifed with mirVana miRNA Isolation Kit (Ambion). PCR was performed for miRs 21-5p, Let7b-5p, 124-3p, and 134-5p. Duplicate CTs were averaged and the relative quantities of miRNAs were calculated using the $2^{-\Delta\Delta CT}$ method and normalized to several artificial spiked-in as controls for extracellular miRNAs.

miR- 21-5p levels in DRG tissue and in macrophages incubated with neuron-derived EVs: Small RNAs were purifed as above and PCR for miR-21-5p performed in L4-L5 DRG obtained from WT and conditional miR-21 KO, DRG after intrathecal oligomers treatment as well as in macrophages after incubation with neuron-derived EVs. Duplicates CTs were averaged and the relative quantities of miRNAs were calculated using the $2^{-\Delta\Delta CT}$ method and normalized to several artificial spiked-in as controls for extracellular miRNAs.

Pro-inflammatory and anti-inflammatory gene markers in transfected macrophages: cDNA was synthesized from total RNA, produced as described above, using the SuperScript II reverse transcriptase kit (Invitrogen™). PCR for Nos2, Mrc1, and Spry2 gene products was performed using a LightCycler FastStart DNA MasterPlus SYBR Green I kit (Roche) in a LightCycler 480 (Roche). Primer sequences are reported in Supplementary Table 3. Duplicate CTs were averaged at the results analyzed by the $2^{-\Delta\Delta CT}$ method using Gapdh or Actb as a housekeeper gene.

**Western blot analysis**. Exosome markers in sensory neurons' media: DRG-derived exosomes were resuspended in cold lysis buffer (Tris-HCl, 20 mM pH 7.5, NaF 10 mM, NaCl 150 mM, 1% Nonidet P-40, phenylmethylsulfonyl fluoride 1 mM, $Na_3VO_4$ 1 mM) containing protease inhibitor cocktail tablets (Roche). Lysates were dissolved in 3× Laemmli's sample buffer, boiled for 5 min, and subjected to 12% SDS-PAGE. Wet transfer was performed using the Bio-Rad Trans-Blot® Cell (Bio-Rad Laboratories, Hertfordshire, UK) for 1 h at 4 °C, and membranes were blocked with 5% non-fat dried milk in TBS-T for 30 min at room temperature. Blots were probed with rabbit anti-TSG101 (1:1000, Cat# T5701, Sigma-Aldrich), rabbit anti-Flotillin-1 (1:1000, Cell Signaling, UK), or rabbit anti-MFG-E8 (1:1000, Cat# sc-33546, Santa Cruz Biotechnology) antibodies. Ponceau red was used as loading control.

Tissue and cell extracts: DRG and macrophages were lysed and protein concentration was determined by Bradford assay (Bio-Rad) prior to gel electrophoresis on 10% SDS-PAGE. Following transfer and blotting, gels were probed with the following primary antibodies: rabbit anti-Spry2 (1:1000; Cell Signaling), rabbit anti-iNOS (1:1000, Cell Signaling), rabbit anti-P-NFκB p65 (1:1000, Cell Signaling), and rabbit anti- NFκB p65 (1:1000, Cell Signaling). α-Tubulin (1:5000) and β-actin (1:1000, Cell Signaling) were used as loading controls. Results were visualized with horseradish peroxidase-coupled anti-rabbit immunoglobulin (Dako) using enhanced chemiluminescence detection reagents. Protein band densities were analyzed by Quantity One (Bio-Rad).

Immunoprecipitation of Dicer in pure DRG neurons: Immunoprecipitation of Dicer from primary neurons was performed by incubating 400 μl of protein lysate with 2 μg of a Dicer antibody (Novus Cat# NBP1-06520). Immunoprecipitates were subjected to SDS-PAGE and immunoblotted with anti-Dicer (1:1000, Novus Cat# NB200-59). Images of the original western blots are presented in Supplementary Fig. 11.

**Flow cytometry analyses**. Isolated DRG neurons and peritoneal macrophages were resuspended in HBSS plus 1.5% BSA. An aliquot of cell suspension was used for counting to derive the absolute number of cells in each sample. Cells were stained on ice for 20 min with anti-mouse CD16/CD32 (Clone 2.4G2, BD Biosciences) to block Fc receptors, followed by incubation with a mix of fluorochrome-conjugated anti-mouse antibodies: CD45.1-Pacific Blue™ (Clone 30-F11, BioLegend), F4/80-PE (Clone BM8, eBioscience), CD11b-APC (Clone M1/70, eBioscience), CD206-PE-Cy7 (Clone C068C2, BioLegend), and CD11c-APC eFluor780® (Clone N418, eBioscience). After washes, cells were diluted in flow buffer and run through a LSRFortessa™ cell analyzer (BD Bioscience). Samples were analyzed with FlowJo software (Tree Start, Ashland, OR, USA).

**Genome-wide microarray analysis of sorted macrophages**. Macrophages (F4/80+ CD11b+ 2–5000 cells) were sorted from a pool of ipsilateral L4/L5 DRG of SNI WT and miR-21 cKO using a FACS Aria II sorter (BD Bioscience). Total RNA was prepared from the cell lysate. Each condition was represented by independently collected biological triplicates. Labeled cell extracts were processed for microarray analysis using the WT Pico Amplification kit (Thermofisher) and hybridized to Affymetrix Mouse 430V2 Arrays. MAS5 pre-processed data were generated in Expression Console (Thermofisher) and analyzed for differential gene expression using Transcript Analysis Console (Thermofisher) with a P value cutoff < 0.05 and two-fold change filter applied. Statistically significant differentially expressed gene list associated with each condition were further annotated and interrogated using MetaCore software (Reuters).

**Statistical analysis**. Statistical analysis was performed with Graph-Pad Prism (Graph-Pad Software). All data are presented as means ± S.E.M. and were analyzed using Student's t test (two groups), one-way ANOVA followed by Bonferroni's multiple comparison test (more than two groups) or two-way ANOVA followed by Tukey test for behavioral data. Differences between means were considered statistically significant when $P < 0.05$. Sample size was chosen to ensure alpha 0.05 and power 0.8. Animal weight was used for randomization and group allocation. No animals were excluded from analysis.

**Data availability**. The authors declare that all data supporting the findings of this study are available from the corresponding author upon request. Microarray data are deposited into Gene Expression Omnibus (GEO) repository system with the accession number GSE104270.

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

## Acknowledgements

This study was supported by European Commission FP7/2007-2013 under grant agreement 602133. We specially thank Biomedical Research Centre (BRC) Flow Cytometry Facility at Guy's, St Thomas' NHS Foundation Trust and King's College London (UK) and Cora Chadick, senior officer Flow Cytometry Facility at EMBL, Monterotondo, Rome. Jonathan Lai and Peter Mouritzen are employees of Exiqon A/S.

## Author contributions

Conceptualization, R.S. and M.M.; methodology, R.S., K.M., V.V., H.A.-A., L.-F.W., T.P., J.G., H.R.J., M.P., J.K., L.C., P.H., and D.C. Investigation, R.S., K.M., L.C., P.H., and M.M.; writing—original draft, M.M.; writing—review and editing, K.M., R.S., M.P., and M.M.; funding acquisition, M.M. and P.H.; resources, T.P., J.L., and P.M.; supervision, L.-F.W., P.H., M.P., and M.M.

## Additional information

**Competing interests:** J.L. and P.M. are employees of Exiqon A/S. The remaining authors declare no competing financial interests.

