## [Peer Review File · Nature Communications]

Reviewers' comments:

Reviewer #1 (Remarks to the Author):

In the present study, Simeoli et al showed that after peripheral nerve injury, DRG sensory neurons secrete extracellular vesicles (EVs), which include exosomes and miRNAs (miR-21), as a mean to communicate with infiltrated macrophages.

First, they showed that nerve injury led to increased miR-21 expression in DRG neurons in vivo. Then by in vitro experiments, they found that capsaicin application increased the release of EVs, which included exosomes and miRNAs (miR-21), from cultured DRG sensory neurons via TRPV1 activation. The isolated capsaicin-induced neuronal exosomes were phagocytosed by cultured peritoneal macrophages, and the isolated EVs affected macrophage polarization. Overexpression of miR-21 in macrophages induced upregulation of pro-inflammatory markers, and favored M1 phenotype. Furthermore, by in vivo experiments, they found intrathecal delivery of miR-21 antagonist or conditional deletion of miR-21 in DRG sensory neurons prevented mechanical hypersensitivity and reduced immune cell recruitment and number of M1 macrophages in DRG following nerve injury.

It is known that miR-21 expression in the DRG neurons is upregulated after peripheral nerve injury, inflammatory macrophages infiltrate the DRG after nerve injury, and miR-21 is a homeostatic regulator of macrophage polarization. This study further demonstrated that the dysregulation of miR-21 in DRG sensory neurons affect inflammatory infiltration after peripheral axon injury, and neuronal EVs act as a plausible mechanism of neuron-macrophage communication after nerve trauma. It is of great interest that neurons can communicate with macrophage via transferring EVs/exosome-containing miR-21. However, to support this novel finding, , more convincing evidence needs to be provided.

Major points

1. In most of the experiments, neuronal EVs were used for the examination of miRs expression and the treatment of macrophages. Therefore, it is improper to conclude "exosomal cargo serves as a mechanism of sensory neuron-macrophage communication".
2. In this study, EVs were isolated by ultracentrifugation at 100,000xg from pure neuron culture media. I am not sure if the authors isolated neuronal EVs just with the method they mentioned in the manuscript or they did not write down the experimental details. This one-step ultracentrifuge will get lots of materials besides EVs, such as cells, dead cells, cell debris, apoptotic bodies, and so on. Therefore, the suspended pellet may lead to many effects on treated cells. The author should pay more attention to the purity of isolated EVs
3. As various contents could also be changed in capsaicin-treated neuronal EVs, it would be better to use EVs isolated from miR-21-overexpressing neurons to treat macrophages. In addition, according to the experimental procedures, isolated DRG neurons were cultured in vitro for 24 hours, stimulated with capsaicin for 3 hours, and then culture media were collected for use. Combining with EVs isolation procedure, it is quite possible that the isolated neuronal EVs were contaminated with capsaicin, which could consequently affect macrophages. Does capsaicin-stimulated TRPV1 KO neuronal EVs affect macrophage polarization?
4. After incubating macrophages with neuronal EVs, why the expression level of miR-21 in macrophages was not significantly increased (see Figure 5b), but the expression of miR-21 target gene *Spry2* was significant reduced (see Figure 4e)?
5. Does neuronal EVs transfer mature or pre-miR-21 into macrophage? Is it possible that the isolated neuronal EVs itself can cause increased transcription of miR-21 in macrophages?

6. Does the inhibition of neuronal EV release in vivo affect macrophage phenotypes and nociceptive hypersensitivity after nerve trauma? Does the reduction of miR-21 in macrophages rescue neuronal EV-induced phenotypes? After nerve trauma, DRG neurons may regulate macrophages via various ways, besides EV-containing miR-21, it would be better to extend the discussion.

6. Whether DRG neurons were cultured in serum-free medium to eliminate bovine serum exosome contamination (Figure 4b)? The detailed materials and procedures should be mentioned in the Methods.

7. Normally, exosomes are isolated by multi-step ultracentrifugation, and then the morphology, diameter distribution, and typical protein expression are examined. In this study, the authors applied western blot and FACS to prove that neuronal EVs contain exosomes. It would be better to examine the morphology and diameter distribution of isolated exosomes. The authors should pay attention to the purity of isolated exosomes, when they study the precise role of neuronal exosomes in neuron-macrophage communication.

9. In Figure 7e, why the N4 control group also showed significant increase of M1 macrophages?

Minor points

1. The representative images in Figure 1a, c and d showed no GFP-positive signals in the control groups. However, according to the quantitative results in Figure 1e, miR-21 fluorescence signal in the control groups was just low, and should be also observed. It would be better to use the original images without adjusting the contrast, or provide other typical images.

2. In Figure 1h and k, the contents of vertical coordinates should be accurately described.

3. The imaging quality should be improved in Figure 1o, and it would be better to show a three-dimensional images of miR-21 and F4/80 signal co-localization. It would be even better to provide the quantitative results of miR-21 intensity in macrophages between groups, since according to in vitro results, miR-21 level should be increased in macrophages after nerve trauma.

4. There are some typos, for example, the "miR-21-5-p" should be "miR-21-5p" and "mi-R21" should be "miR-21" in the Summary and Discussion part.

Reviewer #2 (Remarks to the Author):

This study examined the role miR-21-containing extracellular vesicles (EVs) including exosomes in neuron-macrophage communication and neuropathic pain. The authors showed that spared nerve injury (SNI) caused up-regulation of miR-21 in DRG sensory neurons. They also demonstrated that in DRG neuronal cultures capsaicin induced release of miR-21-containing EVs, which can be taken up by macrophages. Finally, they found that miR-21 changed macrophage phenotype and inhibiting miR-21 or conditional deletion of miR-21 in DRG neurons prevented the development of neuropathic pain. Overall, this is well-conducted multidisciplinary study and the findings are novel and significant. I also have a number of concerns that should be addressed before the paper is accepted for publication.

1. Scrambled probe was used as a control for Fluorescent in situ hybridization (FISH), but the data is not included in Fig.1. It will be helpful to include the negative control result.

2. Fig. 10 is interesting but does not have high resolution. Is it possible to see macrophage uptake of miR-21 using high-resolution images?
3. miRNA levels were quantified by RT-PCR. Is it possible to estimate how many copies of miR-21 a sensory neuron can release after stimulation?
4. miR-21 release was only examined in tissue culture. Will nerve injury cause increase in miR-21 and exosome secretion in CSF?
5. DRG cultures were treated with capsaicin (1 μ M; CAPS) for 3 hours. Will this long-term treatment cause desensitization or neurotoxicity of DRG neurons? Will short-term treatment (<30 min) cause extracellular vesicle (EC) release? Can KCl cause similar EC release as capsaicin?
6. Figure 8 shows that intrathecal delivery of miR-21 antagomir prevents mechanical hypersensitivity after SNI. Can this antagomir treatment also reverse neuropathic pain after post-treatment?
7. Fig. 9: I am curious how many DRGs and how many SNI mice are needed for a single FACS analysis. Only a few DRGs (L3-L6?) are affected by SNI. In contrast, many more DRGs can be collected after chemotherapy treatment.
8. Please include discussion about how miR-21 regulates inflammatory responses in macrophages. Does this require specific mRNA binding? miR-21 may regulate the expression of many other genes including those for anti-inflammatory cytokines.

Reviewer #3 (Remarks to the Author):

This manuscript describes a potentially exciting and provocative mechanism for pain sensitization. The authors have put together a reasonable body of evidence implicating exosomes and particular miR in pain after nerve injury. This story will be of interest to a broad audience of those interested in pain as well as neuro-immuno interactions. A main concern about the paper is that the authors ascribe the effects of the miR-21-5p antagomir to reduce pain behaviors to a site of action in the DRG. It is likely the with 7 days of continuous infusion of the antagomir that this reaches many sites – spinal cord, brain stem, brain, periphery – where it could be acting. An additional important control would be systemic administration of the same dose over the same time period. Also this concern could be addressed in part by determining whether there is an effect of the antagomir in the miR-21 cKO mice. If the authors were to find that the effect of the antagomir were diminished in the cKO mice, this would still not mean that the effect is in the DRG, as the authors conclude, because it is possible that miR-21 products could be released by axons or central terminals of primary afferents (see also below) which would be easily accessible by the intrathecal antagomir.

A second main concern about the paper is the short duration after surgery that the animals were followed. The authors need to determine whether the differences seen in Figures 9 and 10 at day 7 are maintained to day 14 or longer.

Other concerns

In the characterization of the material in the supernatant of the DRG cells, the authors have only looked for a number of markers that may be expressed in exosomes. As such markers would also be expressed in cells how do the authors know that the material is not simply cell fragments or contamination with cells floating in the supernatant? Are there negative controls for markers expressed in cells but not in exosomes?

The authors state in line 128 that exosomes are released by DRG cell bodies. But after time in

culture these cells will likely have substantial processes. How can the authors be sure that the exosomes are not being released from the processes?

In figure 3a there appears to be a very large difference in TSG 101 expression in the CON lane of TRPV1 KO material, and the signal in the CAPS line looks to be increased substantially. How do the authors explain the apparent lack of TSG101 in the TRPV1 KOs?

Minor points

The authors should point the reader to supplementary methods in the appropriate places in the text for, for example, sequence details of reagents when they are first described, methodological details such as incubation times, etc. The main methods are minimalistic.

Reviewers' comments:

Reviewer #1 (Remarks to the Author):

In the present study, Simeoli et al showed that after peripheral nerve injury, DRG sensory neurons secrete extracellular vesicles (EVs), which include exosomes and miRNAs (miR-21), as a mean to communicate with infiltrated macrophages.

First, they showed that nerve injury led to increased miR-21 expression in DRG neurons in vivo. Then by in vitro experiments, they found that capsaicin application increased the release of EVs, which included exosomes and miRNAs (miR-21), from cultured DRG sensory neurons via TRPV1 activation. The isolated capsaicin-induced neuronal exosomes were phagocytosed by cultured peritoneal macrophages, and the isolated EVs affected macrophage polarization. Overexpression of miR-21 in macrophages induced upregulation of pro-inflammatory markers, and favored M1 phenotype. Furthermore, by in vivo experiments, they found intrathecal delivery of miR-21 antagomir or conditional deletion of miR-21 in DRG sensory neurons prevented mechanical hypersensitivity and reduced immune cell recruitment and number of M1 macrophages in DRG following nerve injury.

It is known that miR-21 expression in the DRG neurons is upregulated after peripheral nerve injury, inflammatory macrophages infiltrate the DRG after nerve injury, and miR-21 is a homeostatic regulator of macrophage polarization. This study further demonstrated that the dysregulation of miR-21 in DRG sensory neurons affect inflammatory infiltration after peripheral axon injury, and neuronal EVs act as a plausible mechanism of neuron-macrophage communication after nerve trauma. It is of great interest that neurons can communicate with macrophage via transferring EVs/exosome-containing miR-21. However, to support this novel finding, more convincing evidence needs to be provided.

Major points

1. In most of the experiments, neuronal EVs were used for the examination of miRs expression and the treatment of macrophages. Therefore, it is improper to conclude “exosomal cargo serves as a mechanism of sensory neuron-macrophage communication”.

A new set of experiments has been conducted using NanoSight tracking analysis that determines the size of particles with accuracy, basing data outputs on the Brownian property of small objects. As such, we identified that neuronal-derived vesicles range between 50 and 100 nm in both control and capsaicin samples (Figure 2 panel h and Supplemental Figure 4). Vesicles of this size range are genuinely considered as exosomes. These new data support the Western blot and ImageStream data and suggest that sensory neurons release extracellular vesicles, the large majority of which are exosomes. Therefore we suggest that the term “exosomal cargo” in the title of our manuscript is justified.

2. In this study, EVs were isolated by ultracentrifugation at 100,000xg from pure neuron culture media. I am not sure if the authors isolated neuronal EVs just with the method they mentioned in the manuscript or they did not write down the experimental details. This one-step ultracentrifuge will get lots of materials besides EVs, such as cells, dead cells, cell debris, apoptotic bodies, and so on. Therefore, the suspended pellet may lead to many effects on treated cells. The author should pay more attention to the purity of isolated EVs.

We do apologize for this lack of information and we'd like to thank the reviewer for raising this point. We have now clarified in the supplemental methods that before proceeding to the ultracentrifuge step, supernatants were briefly centrifuged to remove any apoptotic debris or cells, which may contaminate our EVs preparation (Suppl. Material lines 172-173). Furthermore our new NanoSight data (now included in Figure 2h and Supplemental Figure 4) confirm that the large majority of particles are

genuinely exosomes (~90% of events).

3. As various contents could also be changed in capsaicin-treated neuronal EVs, it would be better to use EVs isolated from miR-21-overexpressing neurons to treat macrophages.

We have performed a new set of experiments and over-expressed miR-21 in dissociated DRG by transduction with a lentiviral vector constructed and prepared by our collaborator Liang Wong (reference 24). In these miR-21-overexpressing neurons, we have induced release of EVs by incubation with capsaicin and observed that macrophage uptake of such EVs is associated with increased expression of both miR-21-5p (Fig. 5c) and iNOS (Fig. 5d). When endogenous expression of miR-21-5p was silenced by transfection of macrophages with an antagomir, the incubation of EVs derived from miR-21-overexpressing neurons resulted in significant increase of CD206 (M2 marker) and Spry2 (known miR-21 gene target) (Fig. 5e).

This new set of data obtained with EVs isolated from miR-21-overexpressing neurons confirms and validates our previous observations obtained with normal sensory neurons.

In addition, according to the experimental procedures, isolated DRG neurons were cultured in vitro for 24 hours, stimulated with capsaicin for 3 hours, and then culture media were collected for use. Combining with EVs isolation procedure, it is quite possible that the isolated neuronal EVs were contaminated with capsaicin, which could consequently affect macrophages. Does capsaicin-stimulated TRPV1 KO neuronal EVs affect macrophage polarization?

It is unlikely that capsaicin, if present in the exosomal fraction, would affect macrophages as these cells do not express TRPV1 receptors. Furthermore, capsaicin-containing buffer was replaced with fresh buffer during the exosome isolation procedure as described in the Supplemental Methods section (Suppl. Material lines 175-176).

4. After incubating macrophages with neuronal EVs, why the expression level of miR-21 in macrophages was not significantly increased (see Figure 5b), but the expression of miR-21 target gene Spry2 was significant reduced (see Figure 4e)?

The original Figure 5 b showed increased expression of miR-21 in macrophages after incubation with neuronal EVs. There is a significant ($p < 0.05$) increase of mi-21 in macrophages treated with miR-21 antagomir and EVs compared to macrophages treated with the antagomir only (grey columns).

b

The format of this figure panel (Fig. 5b) has now changed in view of the new set of data that we have included in Figure 5. However, the reported data have not changed from the original submission.

5. Does neuronal EVs transfer mature or pre-miR-21 into macrophage? Is it possible that the isolated neuronal EVs itself can cause increased transcription of miR-21 in macrophages?

These are very interesting points that we are not able to address at the moment. We have inserted these comments in the Discussion lines 316-320.

6A. Does the inhibition of neuronal EV release in vivo affect macrophage phenotypes and nociceptive hypersensitivity after nerve trauma?

This is an interesting question that burdens the extracellular vesicles field: what is the in vivo impact of vesicle generation on a specific biological problem. Unfortunately this issue cannot be addressed with the currently available tools.

The process of vesicle generation is partially known. However whether it varies in a cell-specific manner is certainly unknown. Furthermore, Reviewers would agree that the machinery would vary if one is addressing exosomes versus plasma membrane derived vesicles. In other words, while a legitimate and interesting question, it cannot be addressed with the current scientific knowledge.

However, we are pleased to have been able to perform a microarray analysis of macrophages isolated from WT and cKO DRG at 7 days after SNI. We have identified changes in known miR-21-5p target genes that provide compelling evidence for in vivo relevance of our proposed mechanism.

6B. Does the reduction of miR-21 in macrophages rescue neuronal EV-induced phenotypes?

We have new data showing that treatment with a specific miR-21-5p antagomir prevents neuronal EV-induced increase of iNOS and decrease of Spry2 in macrophages. These data are now reported in figure 4e.

After nerve trauma, DRG neurons may regulate macrophages via various ways, besides EV-containing miR-21, it would be better to extend the discussion.

We have included a sentence about sensory neuron-derived peptides that mediate neurogenic inflammation (Lines 324-325).

6C. Whether DRG neurons were cultured in serum-free medium to eliminate bovine serum exosome contamination (Figure 4b)? The detailed materials and procedures should be mentioned in the Methods.

We have clarified this point in the supplemental methods line 162.

7. Normally, exosomes are isolated by multi-step ultracentrifugation, and then the morphology, diameter distribution, and typical protein expression are examined. In this study, the authors applied western blot and FACS to prove that neuronal EVs contain exosomes. It would be better to examine the morphology and diameter distribution of isolated exosomes. The authors should pay attention to the purity of isolated exosomes, when they study the precise role of neuronal exosomes in neuron-macrophage communication.

To address this point and check for purity and presence of exosomes in our EVs preparation, we have analysed EVs using a NanoSight (NS300, Malvern). According to the data obtained (now reported in Figure 2h and Supplemental Figure 4a), we can clearly conclude that our samples are rich in particles, 90% of which are below ~ 125nm in diameter, i.e. highly enriched in exosomes.

9. In Figure 7e, why the N4 control group also showed significant increase of M1 macrophages?

We do apologize for this error. We have checked this set of data and performed new statistical analysis using raw data. Panel 7e has been update in the revised version.

Minor points

1. The representative images in Figure 1a, c and d showed no GFP-positive signals in the control groups. However, according to the quantitative results in Figure 1e, miR-21 fluorescence signal in the control groups was just low, and should be also observed. It would be better to use the original images without adjusting the contrast, or provide other typical images.

The data presented in Figure 1e are representative of an average of all of the sections quantified and indeed in some cases there was no detectable fluorescence (without changing the acquisition settings). We have replaced the image in panel d with one that reflects low fluorescence as opposed to no fluorescence.

2. In Figure 1h and k, the contents of vertical coordinates should be accurately described.

In Figure 1 legend we have highlighted in red font the description of the y-axes labels for panels h and k that we hope clarifies their significance.

3. The imaging quality should be improved in Figure 1o, and it would be better to show a three-dimensional images of miR-21 and F4/80 signal co-localization. It would be even better to provide the quantitative results of miR-21 intensity in macrophages between groups, since according to in vitro results, miR-21 level should be increased in macrophages after nerve trauma.

We have inserted a z-stack confocal higher-resolution image in panel 1o that better captures the presence of miR-21 in F4/80 positive cells.

Furthermore, we have quantified miR-21 positive cells that co-localize with F4/80 in ipsilateral L4 and L5 DRG of SNI and SHAM injured mice. The quantification is now reported in supplemental Fig. 1e.

4. There are some typos, for example, the “miR-21-5-p” should be “miR-21-5p” and “mi-R21” should be “miR-21” in the Summary and Discussion part.

We have check-proofed our manuscript.

Reviewer #2 (Remarks to the Author):

This study examined the role miR-21-containing extracellular vesicles (EVs) including exosomes in neuron-macrophage communication and neuropathic pain. The authors showed that spared nerve injury (SNI) caused up-regulation of miR-21 in DRG sensory neurons. They also demonstrated that in DRG neuronal cultures capsaicin induced release of miR-21-containing EVs, which can be taken up by macrophages. Finally, they found that miR-21 changed macrophage phenotype and inhibiting miR-21 or conditional deletion of miR-21 in DRG neurons prevented the development of neuropathic pain. Overall, this is well-conducted multidisciplinary study and the findings are novel and significant. I also have a number of concerns that should be addressed before the paper is accepted for publication.

1. Scrambled probe was used as a control for Fluorescent in situ hybridization (FISH), but the data is not included in Fig.1. It will be helpful to include the negative control result.

We have included representative images for the scrambled probe in Supplementary Fig. 1a-d.

2. Fig. 1o is interesting but does not have high resolution. Is it possible to see macrophage uptake of miR-21 using high-resolution images?

The image used in panel 1o has been changed and it now shows a z-stack confocal higher-resolution image, which we feel better captures the co-localisation between miR-21 and F4/80.

3. miRNA levels were quantified by RT-PCR. Is it possible to estimate how many copies of miR-21 a sensory neuron can release after stimulation?

We have attempted to quantify miR-21-5p in the exosomal fraction of cultured DRG media using the hsa-miR-21-5p miREIA by BioVendor (Cat No.: RDM0001H). This kit has just become commercially available. Unfortunately, sample concentrations of miR-21-5p were below the detection limit of the assay (0.20 amol/ul) and we are unable to add this type of quantification in our manuscript. We have confirmed the presence of miR-21-5p by RT-PCR in the same samples that were run through the miREIA.

4. miR-21 release was only examined in tissue culture. Will nerve injury cause increase in miR-21 and exosome secretion in CSF?

We cannot address this interesting question as it is not feasible to obtain CSF from mice. Even if we were to implement a procedure to do so, the very small volume withdrawn from a single mouse would require pooling samples from several mice thereby questioning the ethical value of such experiment.

5. DRG cultures were treated with capsaicin (1 μ M; CAPS) for 3 hours. Will this long-term treatment cause desensitization or neurotoxicity of DRG neurons? Will short-term treatment (<30 min) cause extracellular vesicle (EC) release? Can KCl cause similar EC release as capsaicin?

1. In order to check the cell viability following 3 hours incubation with capsaicin, we have performed a Pierce LDH cytotoxicity assay and observed no signs of neurotoxicity (less than 1% compared to control vehicle). We have included these observations as data not shown in line 109.

2. We have tested the effect of a 25 min-incubation of capsaicin and observed significant increase of TSG101 but not MFG-E8 and Flotillin-1. These new data are now reported in Supplemental Figure 2.

3. Incubation of sensory neurons in culture with KCl for 3 hour at 25 and 50 mM was associated with significant increase in expression of the exosomal markers TSG101 and MFG-E8. However, 50mM was required for the detection of Flotillin-1 band. These new data are reported in Supplemental Figure 3.

Altogether these data indicate that the optimal conditions for the release of our selected 3 exosomal markers require 3 hours incubation of capsaicin 1 uM and KCl 50 mM.

In this context, we wish to highlight that alpha-tubulin did not appear to be an accurate control for Western blot of exosomal markers as levels varied between loaded samples. Thus, we have used Ponceau red as loading control that enables the determination of equal protein loading on filters. Indeed, others have used Ponceau red as loading control for MFG-E8 in cell media (Reference 28). For this reason, we have now used Ponceau red as loading control in Figures 2, 3 Suppl. Figures 2 and 3.

6. Figure 8 shows that intrathecal delivery of miR-21 antagomir prevents mechanical hypersensitivity after SNI. Can this antagomir treatment also reverse neuropathic pain after post-treatment?

We agree that it would be very interesting to determine whether the antagomir reverses established allodynia. However, this set of experiments will be performed in future studies in which we will also determine whether miR-21 up-regulation in DRG is significant 2 weeks after nerve injury.

7. Fig. 9: I am curious how many DRGs and how many SNI mice are needed for a single FACS analysis. Only a few DRGs (L3-L6?) are affected by SNI. In contrast, many more DRGs can be collected after chemotherapy treatment.

We pool L4 and L5 DRG from 6 mice, so a total of 12 DRG for one replicate. Details have been included in the Supplemental Methods section line 455.

8. Please include discussion about how miR-21 regulates inflammatory responses in macrophages. Does this require specific mRNA binding? miR-21 may regulate the expression of many other genes including those for anti-inflammatory cytokines.

We acknowledge evidence that miR-21 may upregulate IL-10 in macrophages after 24 hours-TLR4 stimulation which mimics an infective status (lines 348-350).

In addition, we have included a sentence about the possibility that miR-21-5p may regulate the expression of genes that regulate cytoskeleton remodelling and cell survival (367-370). This suggestion derives from the microarray analysis performed in macrophages obtained from WT and cKO DRG and reported in supplemental Figure 10.

Reviewer #3 (Remarks to the Author):

This manuscript describes a potentially exciting and provocative mechanism for pain sensitization. The authors have put together a reasonable body of evidence implicating exosomes and particular miR in pain after nerve injury. This story will be of interest to a broad audience of those interested in pain as well as neuro-immuno interactions. A main concern about the paper is that the authors ascribe the effects of the miR-21-5p antagomir to reduce pain behaviors to a site of action in the DRG. It is likely the with 7 days of continuous infusion of the antagomir that this reaches many sites – spinal cord, brain stem, brain, periphery – where it could be acting.

We agree that we cannot rule out the possibility that the intrathecal antagomir might have acted at the peripheral terminals of sensory neurons as we have not checked for either miR-21 expression or antagomir presence in the paw skin. However, we provide evidence that intrathecal administration of the antagomir is not associated with fluorescence the spinal cord (suppl. Fig. 7). Therefore, the antagomir is not likely to have accessed to the central terminals of primary afferent fibres. Also we think it unlikely that this oligomer would reach centres higher than the cord at effective concentrations (lines 365-366).

We have focussed on the DRG because sensory neuron cell bodies can form multivesicular bodies and release exosomes, whilst sensory axons do not share this property (reference 12). We do not wish to claim that miR-21 and its products are exclusively located in the DRG. Indeed, other members of our Consortium are investigating possible roles of miR-21 in the peripheral nerves and the brain.

We agree with this Referee that miR-21 products may be released from the central and peripheral terminals of primary afferent fibres and have included a sentence in Discussion in lines 343-344.

An additional important control would be systemic administration of the same dose over the same time period. Also this concern could be addressed in part by determining whether there is an effect of the antagomir in the miR-21 cKO mice. If the authors were to find that the effect of the antagomir were diminished in the cKO mice, this would still not mean that the effect is in the DRG, as the authors conclude, because it is possible that miR-21 products could be released by axons or central terminals of primary afferents (see also below) which would be easily accessible by the intrathecal antagomir.

We have performed a new experiment and delivered systemically the same dose of antagomir delivered intrathecally, as asked by this Referee. We observed no anti-allodynic effect and no decrease in number of infiltrating macrophages (supplemental Figure 9). A plausible explanation for the lack of efficacy of the systemic delivery of the oligomer is that the dose used did not reach sufficient concentration to inhibit miR-21 in peripheral targets including DRG and peripheral terminals of sensory neurons in the skin (Results lines 249 and 252 and in Discussion lines 301-306).

A second main concern about the paper is the short duration after surgery that the animals were followed. The authors need to determine whether the differences seen in Figures 9 and 10 at day 7 are maintained to day 14 or longer.

We agree that it would be of great interest to determine the effect of both antagomir and conditional miR-21 ko on established neuropathic allodynia and macrophage phenotype 2 weeks after nerve injury. However, these experiments will be part of our future studies focussed on miR-21 as a target for pain

Other concerns

In the characterization of the material in the supernatant of the DRG cells, the authors have only looked for a number of markers that may be expressed in exosomes. As such markers would also be expressed in cells how do the authors know that the material is not simply cell fragments or contamination with cells floating in the supernatant? Are there negative controls for markers expressed in cells but not in exosomes?

We do apologize for this lack of information and we'd like to thank the reviewer for raising this point. We have now clarified in the supplemental methods that before proceeding to the ultracentrifuge step, supernatants were briefly centrifuged to remove any apoptotic debris or cells which may contaminate our EVs preparation (suppl. Material lines 172-173). Furthermore new Nanosight data (now included in Figure 2h and Suppl. Fig. 4a) confirm the quality as well as the purity (>90%) of our exosomes preparation.

The authors state in line 128 that exosomes are released by DRG cell bodies. But after time in culture these cells will likely have substantial processes. How can the authors be sure that the exosomes are not being released from the processes?

We maintain DRG neurons in culture for 24 hours and they do not show processes at this time point.

In figure 3a there appears to be a very large difference in TSG 101 expression in the CON lane of TRPV1 KO material, and the signal in the CAPS line looks to be increased substantially. How do the authors explain the apparent lack of TSG101 in the TRPV1 KOs?

A new representative Western blot image for TSG 101 has been included in figure 3a.

Minor points

The authors should point the reader to supplementary methods in the appropriate places in the text for, for example, sequence details of reagents when they are first described, methodological details such as incubation times, etc. The main methods are minimalistic.

We have included details such as incubation times in the Results section and the figure legends. Unfortunately, we had to shorten the main methods in order to include new results and discussion of the new data. We hope the Supplemental methods is sufficiently detailed.

Reviewers' comments:

Reviewer #1 (Remarks to the Author):

Most of my concerns have been well addressed in the revised manuscript. I have no more concern now.

Reviewer #2 (Remarks to the Author):

The authors did a great job to address my concerns. This is an important contribution to the field. The authors should be congratulated. In addition to let-7b, this is another example of miRNA release by neuronal activity. This study provides a novel interaction of sensory neurons and macrophages underlying the pathogenesis of pain.

Reviewer #3 (Remarks to the Author):

In the revised manuscript the authors have done some additional experiments and made clarifications that improve the paper. It is disappointing that they have refused to include new data where they follow the animals for longer periods to determine whether the effects seen in Figs. 9 and 10 persist. Such data would have even more significantly improved the manuscript.

A main concern I have with the revised manuscript is that Supplementary Fig 7 is not at all convincing. The images are completely black with dashed curves drawn on. For such negative results there needs to be some positive detection within the image – fiduciary marks or something – to indicate that the authors would be able to detect a signal if it were there. Also, I would suggest a lower power image so that the reader could see whether this is indeed lumbar spinal cord. Is there signal on the surface of the cord or pial membranes, as might be expected (given the degree of signal in the DRGs) if there was intrathecal injection and a relevant section of spinal cord was imaged? As is this figure is unacceptable.

Reviewers' comments:

Reviewer #1 (Remarks to the Author):

Most of my concerns have been well addressed in the revised manuscript. I have no more concern now.

Reviewer #2 (Remarks to the Author):

The authors did a great job to address my concerns. This is an important contribution to the field. The authors should be congratulated. In addition to let-7b, this is another example of miRNA release by neuronal activity. This study provides a novel interaction of sensory neurons and macrophages underlying the pathogenesis of pain.

Reviewer #3 (Remarks to the Author):

In the revised manuscript the authors have done some additional experiments and made clarifications that improve the paper. It is disappointing that they have refused to include new data where they follow the animals for longer periods to determine whether the effects seen in Figs. 9 and 10 persist. Such data would have even more significantly improved the manuscript.

A main concern I have with the revised manuscript is that Supplementary Fig 7 is not at all convincing. The images are completely black with dashed curves drawn on. For such negative results there needs to be some positive detection within the image – fiduciary marks or something – to indicate that the authors would be able to detect a signal if it were there. Also, I would suggest a lower power image so that the reader could see whether this is indeed lumbar spinal cord. Is there signal on the surface of the cord or pial membranes, as might be expected (given the degree of signal in the DRGs) if there was intrathecal injection and a relevant section of spinal cord was imaged? As is this figure is unacceptable.

Apologies for the quality of the images used the suppl. fig 7 in which we show that there is no fluorescence in the dorsal horn. We have provided new images in which there is no positive oligomer fluorescence yet the dorsal horn is easily identifiable.

REVIEWERS' COMMENTS:

Reviewer #3 (Remarks to the Author):

The authors have addressed my concern.